# Sequential Signal Mixing Aggregation for Message Passing Graph Neural Networks

**Mitchell Keren Taraday**[*]
Department of Computer Science
Technion
Haifa, Israel
`butovsky.mitchell@gmail.com`

**Almog David**[*]
Department of Computer Science
Technion
Haifa, Israel
`almogdavid@gmail.com`

**Chaim Baskin**
School of Electrical and Computer Engineering
Ben-Gurion University of the Negev
Be'er Sheva, Israel
`chaimbaskin@bgu.ac.il`

## Abstract

Message Passing Graph Neural Networks (MPGNNs) have emerged as the preferred method for modeling complex interactions across diverse graph entities. While the theory of such models is well understood, their aggregation module has not received sufficient attention. Sum-based aggregators have solid theoretical foundations regarding their separation capabilities. However, practitioners often prefer using more complex aggregations and mixtures of diverse aggregations. In this work, we unveil a possible explanation for this gap. We claim that sum-based aggregators fail to "mix" features belonging to distinct neighbors, preventing them from succeeding at downstream tasks. To this end, we introduce Sequential Signal Mixing Aggregation (SSMA), a novel plug-and-play aggregation for MPGNNs. SSMA treats the neighbor features as 2D discrete signals and sequentially convolves them, inherently enhancing the ability to mix features attributed to distinct neighbors. By performing extensive experiments, we show that when combining SSMA with well-established MPGNN architectures, we achieve substantial performance gains across various benchmarks, achieving new state-of-the-art results in many settings. We published our code at `https://almogdavid.github.io/SSMA/`

## 1  Introduction

Message-passing Graph Neural Networks (MPGNNs) have established themselves as the major workhorses for graph representation learning over the past decade [24]. These models have been proven to be effective in graph-structured problems in a variety of domains, ranging from social networks [18] to natural sciences [14, 23, 4] and having some non-trivial applications in computer vision and natural language processing [26, 33, 45, 28].

Such renowned models of this nature owe their success to their high efficiency, along with good generalization capabilities and simplicity. A typical MPGNN takes graph-structured data containing node and edge features as input. It then iteratively updates node representations by combining their egocentric view with a symmetrized aggregation of their proximate neighbor features.

---

[*]Equal contribution.

38th Conference on Neural Information Processing Systems (NeurIPS 2024).

The key insight regarding the expressive power of such models is their equivalence to the Weisfeller-Lehman (WL) graph isomorphism test [47]. Consequently, past research directions were majorly directed toward developing models that surpass the vanilla WL test by tackling the graph learnability problem from various perspectives, including stronger notions of the WL test [31, 35], spectral graph methods [46, 13, 44] and graph transformers [49, 36].

However, one subtle but often overlooked detail in such expressivity analyses is the existence of a Hash function, which compresses the neighbor features into a fixed-sized representation. Such Hash function need not only be injective but also differentiable and efficient in terms of memory

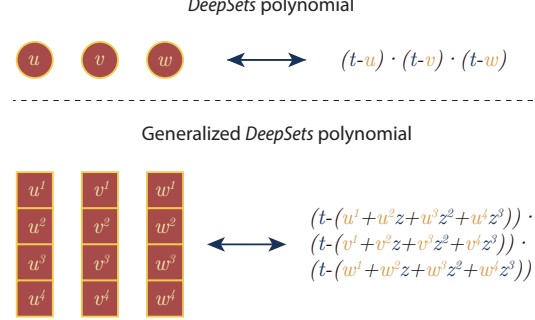

Figure 1: An efficient and provable generalization of the *DeepSets* polynomial to vector features.

and computation. The seminal DeepSets paper [50] showcased such a sum-based construction for the Hash function. While this construction was very simple and computationally efficient, the theoretical representation size required in this construction is exponential in the node feature dimension. Although the bound on this representation size was improved in later works [17, 1], sum-based aggregations seem to lag behind the aggregators used in practice [11].

In this work, we suggest that a possible explanation for this gap is the inability of sum-based aggregators to "mix" features belonging to distinct neighbors. We formalize the "neighbor-mixing" property and show that sum-based aggregators have limited neighbor-mixing capability. This observation is later verified by conducting an experiment showing that sum-based aggregators struggle with approximating even a very simple function requiring neighbor-mixing.

With this motivation in mind, we propose a new aggregation module that treats the neighbor features as two-dimensional discrete signals and sequentially convolves them - hence coined as Sequential Signal Mixing Aggregation (SSMA). SSMA has a provably polynomial representation size $m = \mathcal{O}(n^2d)$ (where $n$ is the number of neighbors and $d$ is feature dimensionality). The theoretical construction underlying SSMA provides a positive answer to a lasting mystery regarding DeepSets [50] - "Can the *DeepSets* polynomial be **efficiently** generalized to handle vector features?"as depicted in Figure 1.

As later investigated, the convolutional component in SSMA allows it to directly mix features attributed to distinct neighbors, inducing a higher-order notion of neighbor mixing. We then discuss some practical aspects of SSMA. Particularly, we discuss how to implement it in a computationally efficient manner, how to scale it to larger graphs and how to make it easier to optimize.

Finally, we demonstrate that when integrated into a wide range of well established MPGNN architectures, SSMA greatly enhances their performance. We observe significant gains across all benchmarks tested, including the TU datasets [32], open graph benchmark (OGB) [21] datasets, long-range graph benchmarks (LRGB) [16] datasets and the ZINC [19] molecular property prediction dataset achieving state-of-the-art results in many settings.

**Contributions.**  Our contributions may be summarized as follows:

1. We define the notion of "neighbor-mixing" and show that sum-based aggregators have limited neighbor-mixing power. We verify this idea by conducting an experiment on a simple and natural synthetic task.

2. We propose Sequential Signal Mixing Aggregation (SSMA) - an aggregation module of dimension $m = \mathcal{O}(n^2d)$ which treats the neighbor features as discrete signals and sequentially convolves them. The theoretical construction underlying SSMA builds upon the *DeepSets* polynomial, efficiently extending it to multidimensional features.

3. We introduce a few practices for stabilizing the optimization process of SSMA and show how to scale it to larger graphs.

4. Finally, we conduct extensive experiments showing that enriching prominent MPGNN architectures with SSMA yields large improvements on a variety of benchmarks, achieving state-of-the-art results.

## 2 Preliminaries and related work

Let $\mathcal{X}$ be some domain. We are interested in representing **multisets** (sets in which repeated elements are allowed) over that domain. We denote multisets by $\{\{x_1, ..., x_n\}\}$ where each $x_i \in \mathcal{X}$, and denote by $\mathcal{M}_n := (\mathcal{X})^n$ the $n$-tuple space over $\mathcal{X}$. We seek a (possibly learnable) permutation invariant mapping $f : \mathcal{M}_n \to \mathbb{R}^m$ separating distinct multisets [2]. When combined with a learnable compression network $g_\theta : \mathbb{R}^m \to \mathcal{X}$, their composition $\gamma = g_\theta \circ f$ can be utilized as an aggregation module for MPGNNs over the domain $\mathcal{X}$.

Particularly, we are interested in continuous features, namely the domains $\mathcal{X} = \mathbb{R}$ and $\mathcal{X} = \mathbb{R}^d$. We consider the symmetry group $\mathcal{G} = S_n$ acting on $\mathcal{M}_n = \mathbb{R}^n$ by $[\sigma.\boldsymbol{x}]_i = \boldsymbol{x}_{\sigma^{-1}(i)}$ and on $\mathcal{M}_n = \mathbb{R}^{n \times d}$ by $[\sigma.\mathbf{X}]_{ij} = \mathbf{X}_{\sigma^{-1}(i)j}$ correspondingly. It is widely agreed that finding a good representation $f : \mathcal{M}_n \to \mathbb{R}^m$ for these domains is crucial for building better aggregation modules $\gamma$ and has a direct influence on the performance of the model on a variety of downstream tasks [47, 12, 27, 40].

DeepSets [50] was the pioneering work introducing a sum-based aggregator with a provably finite representation size $m$: $\gamma(\{\{x_1, ..., x_n\}\}) = \rho(\sum_{k=1}^n \phi(x_k))$ where $\phi : \mathbb{R}^d \to \mathbb{R}^m$ and $\rho : \mathbb{R}^m \to \mathbb{R}^d$. Their construction consisted of "hand-crafted" moment-based features. Despite being efficient for scalar-based features, the representation size grew exponentially with the node feature dimensionality, $m \in \mathcal{O}(\binom{n+d}{d})$. This upper bound was later improved to $\mathcal{O}(n^2d)$ and eventually to a tight $\Theta(nd)$ [17]. While moment-based features served as a powerful tool for achieving theoretical separation, learnable neural features are favored over such hand-crafted features in practice. As was unveiled, neural features can achieve theoretical separation as well, as long as non-polynomial analytic activations are used [1].

Despite their clear theoretical advantages, sum-based aggregators seem to have limited performance in practice [12, 27]. Consequently, many works focused on different species of permutation invariant aggregators. For instance, attention-based aggregators have been proposed to capture the most important signals incoming from the neighborhood [2, 7]. Others suggested using a mixture of symmetric aggregators such as min, max, mean, sum, std as each of these aggregators helps separate different kinds of multisets [47, 41, 12]. Other works focused on aggregations preserving intrinsic properties of the neighborhood data such as variance and fisher-information [38, 30].

Another intriguing type of work deals with the relaxation of the neighbor ordering invariance constraint. Particularly, regularizing recurrent neural network-based aggregations to maintain permutation invariance - either by choosing a random neighbor permutation [20] or by explicit regularization terms [10, 34] has raised some interest.

## 3 On the limited neighbor-mixing of sum-based aggregators

Despite their provable separation power, sum-based aggregators seem to lag behind other aggregators used in practice [11]. We claim that a possible explanation for this phenomenon lies in their inability to "mix" the neighbor's features, in that the mutual effect of perturbing the features of two distinct neighbors on each aggregation output is very small. In practice, many downstream tasks require high "mixing" values as the aggregator should mix information from different distinct neighbors to produce a useful representation for tackling the downstream task.

**Definition 3.1.** Let $\gamma : \mathbb{R}^{n \times d} \to \mathbb{R}^d$ be some aggregation function that is continuously twice differentiable. We define the *neighbor mixing* of the $\ell$-th aggregation output with respect to the neighbor pair $(i, j)$:

$$\text{mix}_{i,j}^{(\ell)} := \left\| \frac{\partial^2}{\partial x_i \partial x_j} \gamma^{(\ell)}(x_1, ..., x_n) \right\|_2 \tag{1}$$

At an intuitive level, sum-based aggregators have small $\text{mix}_{i,j}^{(\ell)}$ values as the result of the local pooling operation is summed across the neighbors. Namely, without explicitly "mixing" features from distinct neighbors before the summation. Indeed, given $\gamma(\{\{x_1, ..., x_n\}\}) = \sum_{k=1}^n \phi(x_k)$ we have:

---

[2] meaning that distinct multisets should be mapped to distinct representations, hereby requiring a sufficiently large representation dimension $m$.

$$\frac{\partial^2}{\partial x_i \partial x_j} \sum_{k=1}^{n} \phi^{(\ell)}(x_k) = 0 \tag{2}$$

Formally, to account for mixing that may occur in any subsequent (global) transformation we have the following proposition:

**Proposition 3.2.** *Let* $\gamma(\{\{x_1, ..., x_n\}\}) = \rho\left(\sum_{k=1}^{n} \phi(x_k)\right)$ *where* $\phi : \mathbb{R}^d \to \mathbb{R}^m$ *is a local operator and* $\rho : \mathbb{R}^m \to \mathbb{R}^d$ *is a pooling operator that is continuously twice differentiable. Then, we have* $\forall i \neq j$:

$$\mathsf{mix}_{i,j}^{(\ell)} \leq \|J_\phi(x_i)\|_2 \cdot \left\|H_{\rho^{(\ell)}}(\sum_{k=1}^{n} \phi(x_k))\right\|_2 \cdot \|J_\phi(x_j)\|_2 \tag{3}$$

*Where* $J_\phi(.)$ *is the Jacobian matrix of* $\phi$ *and* $H_{\rho^{(\ell)}}(.)$ *is the Hessian matrix corresponding to* $\ell$-*th output of* $\rho$. *Particularly, for typical choices of* $\phi$ *and* $\rho$ *it follows:* $\mathsf{mix}_{i,j}^{(\ell)} \in \mathcal{O}(\|\theta\|_2^2)$ *where* $\theta$ *is the concatenation of the parameters in* $\phi$ *and* $\rho$.

The proof of Proposition 3.2 is given in Appendix A.1.

Motivated by the above observation, we propose a new species of aggregation module which is convolution-based rather than sum-based.

## 4    SSMA - Sequential Signal Mixing Aggregation

### 4.1    Warm-up: *DeepSets* polynomial from a *convolutional* point of view

Let $\overline{\boldsymbol{x}} = \{\{\boldsymbol{x}_1, ..., \boldsymbol{x}_n\}\}$ be a scalar multiset. We define its *DeepSets* polynomial by considering a polynomial of variable $t$ having the multiset elements as its roots:

$$p_{\overline{\boldsymbol{x}}}(t) := \prod_{i=1}^{n}(t - \boldsymbol{x}_i) \tag{4}$$

Its coefficients, we denote by $e_k(\boldsymbol{x})$, are permutation invariant functions. Moreover, the $(e_k(\boldsymbol{x}))_{k=0}^{m}$s form an ensemble of invariant separators [3].

Instead of describing a polynomial by its coefficients, one can represent a polynomial by evaluating it on some fixed set of points. Given a set of $n+1$ fixed points, the polynomial may be represented by evaluating its value on these points. One can switch from this representation back to the coefficients by solving a system of linear equations, which always has a unique solution. Now, by allowing the evaluation points to be complex, we can choose them as the roots of unity. By doing so, we get the discrete Fourier transform (DFT) of the polynomial coefficients:

$$\boldsymbol{\zeta}_j(\boldsymbol{x}) = \sum_{k=0}^{n} e_k(\boldsymbol{x}) \cdot e^{-\frac{2\pi i j}{n+1}k} \quad (j = 0, ..., n) \tag{5}$$

Next, we denote the factors in $p_{\overline{\boldsymbol{x}}}(t) = \prod_{i=1}^{n} p_i(t)$ where $p_i(t) := t - \boldsymbol{x}_i$. The (padded) coefficients of each $p_i(t)$ are then given by the affine transformation:

$$\boldsymbol{h}(\boldsymbol{x}_i) = [-\boldsymbol{x}_i, 1, 0, ..., 0] \in \mathbb{R}^{n+1} \tag{6}$$

The nice thing about representing a polynomial by evaluating its values at a list of fixed points is that polynomial multiplication becomes *point-wise*. It can be deduced that the coefficients of $p_{\overline{\boldsymbol{x}}}(t)$ can be computed by transforming the coefficients $\boldsymbol{h}(\boldsymbol{x}_i)$ of each $p_i(t)$ to the Fourier domain, performing

---

[3]If for two multisets we have $e_k(\boldsymbol{x}) = e_k(\boldsymbol{y})$ for all $0 \leq k \leq m$, then $p_{\overline{\boldsymbol{x}}}(t) = p_{\overline{\boldsymbol{y}}}(t)$ for all $t$, and therefore each one of the roots of the left-hand side polynomial corresponds to some root of the right-hand side polynomial. An inductive argument shows that this implies that both polynomials have the same roots.

elementwise multiplication and then transforming back to the coefficients domain. According to the circular convolution theorem, this exactly amounts to sequentially convolving the coefficients $h(x_i)$. We combine the above ideas into the following theorem:

**Theorem 4.1.** *Scalar multisets* $\overline{x} = \{\{x_1, ..., x_n\}\}$ *can be represented by an invariant and separating map* $f_{conv}$:

$$f_{conv}(x) = \underset{i=1}{\overset{n}{\circledast}} \, h(x_i) \tag{7}$$

*Where* $h : \mathbb{R} \to \mathbb{R}^m$ *is an affine map,* $\circledast$ *is the circular convolution operator, and the number of separators is* $m = n + 1 \in \mathcal{O}(n)$.

Theorem 4.1 simply states that sequential convolution can be utilized to compute the renowned *DeepSets* polynomial coefficients. While not particularly surprising, Theorem 4.1 shows that the coefficients of the *DeepSets* polynomial can be efficiently computed and directly utilized as a multiset representation. Moreover, it paves the way for our construction, as seen in the next section.

## 4.2 Efficient generalization to multidimensional features

"How does the *DeepSets* polynomial can be **efficiently** extended to handle vector features?"

The key idea underlying our answer to this question is to encode each feature vector as another polynomial, and then to reduce the problem to the scalar case.

---

**Generalized DeepSets Polynomial**

We encode each element $\mathbf{X}_i$ belonging to the multiset $\overline{\mathbf{X}} = \{\{\mathbf{X}_1, ..., \mathbf{X}_n\}\}$ as a polynomial of *another* variable $z$:

$$\mathsf{Enc}(\mathbf{X}_i) = \sum_{j=1}^{d} \mathbf{X}_{ij} \cdot z^{j-1} \tag{8}$$

Then, we can perform a reduction to the scalar case by replacing each $\mathbf{X}_i$ with $\mathsf{Enc}(\mathbf{X}_i)$:

$$p_i(t, z) := t - \mathsf{Enc}(\mathbf{X}_i) = t - \sum_{j=1}^{d} \mathbf{X}_{ij} \cdot z^{j-1} \tag{9}$$

And define the generalized *DeepSets* polynomial:

$$p_{\overline{\mathbf{X}}}(t, z) := \prod_{i=1}^{n} p_i(t, z) = \sum_{k,l} e_{k\ell}(\mathbf{X}) \cdot t^k z^\ell \tag{10}$$

Where $e_{k\ell}(\mathbf{X})$ is the coefficient of $t^k z^\ell$ in $p_{\overline{\mathbf{X}}}(t, z)$. Note $0 \le k \le n$ while $0 \le \ell \le n(d-1)$.

---

Opposed to the scalar case, it is not evident why the obtained coefficients $(e_{k\ell}(\mathbf{X}))_{k,\ell}$ in the above construction form an ensemble of separators. We prove injectivity by utilizing ideas from ring theory, particularly the notions of unique factorization domains (UFDs) and Gauss's lemma in Appendix A.2.

We can now repeat the steps in Section 4.1 to achieve the actual representation. We compute the coefficient *matrix* of each $p_i(t, z)$ and sequentially perform two-dimensional circular convolution.

This leads us to an analogous theorem for the $d$-dimensional case:

**Theorem 4.2.** *Vector multisets* $\overline{\mathbf{X}} = \{\{\mathbf{X}_1, ..., \mathbf{X}_n\}\}$ *can be represented by an invariant and separating map* $f_{conv}$:

$$f_{conv}(\mathbf{X}) = \underset{i=1}{\overset{n}{\circledast}} \, \Phi(\mathbf{X}_i) \tag{11}$$

*Where* $\Phi : \mathbb{R}^d \to \mathbb{R}^{m_1 \times m_2}$ *is an affine map,* $\circledast$ *is the 2D circular convolution operator and the number of separators is* $m = m_1 \times m_2 = (n+1)(n(d-1)+1) \in \mathcal{O}(n^2 d)$.

The full proof of Theorem 4.2 is given in Appendix A.3.

$$\frac{\partial^2 h^1}{\partial u^0 \partial v^1} = 1 \qquad\qquad \frac{\partial^3 h^0}{\partial u^2 \partial v^1 \partial w^1} = 1$$

Figure 2: Visualization of the higher order notion of neighbor mixing. We visualize the convolution result $h$ for 3-dimensional features, considering 2 neighbors $u, v$ (left) and 3 neighbors $u, v, w$ (right). We demonstrate for each $n$-tuple matching a feature per node, the corresponding $n$-th order derivative of exactly one entry of $h$ is 1.

### 4.3 How does circular convolution impact neighbor Mixing?

Let $\boldsymbol{u}_1, ..., \boldsymbol{u}_n \in \mathbb{R}^m$ be discrete signals representing the locally-transformed neighbors before being aggregated. For the sake of simplicity, we slightly override the notation in this section, and refer to the $j$-th element of the $i$-th signal as $\boldsymbol{u}_i^j$ with $j$ starting from 0.

The core factor causing the neighbor mixing bottleneck of sum-based aggregators $\boldsymbol{h} = \sum_{i=1}^n \boldsymbol{u}_i$ lies within the fact that no mixing is done in the representation, but only in the MLP compressor that comes afterward:

$$\frac{\partial^2}{\partial \boldsymbol{u}_i^k \partial \boldsymbol{u}_j^\ell} \boldsymbol{h} = 0 \tag{12}$$

On the contrary, each element of sequential circular convolution $\boldsymbol{h} = \boldsymbol{u}_1 \circledast ... \circledast \boldsymbol{u}_n$ is composed of sums of terms of the form $\boldsymbol{u}_1^{j_1} \boldsymbol{u}_2^{j_2} \cdot ... \cdot \boldsymbol{u}_n^{j_n}$. Particularly:

$$\boldsymbol{h}^k = \sum_{\substack{j_1 + ... + j_n \equiv k \\ (\bmod m)}} \boldsymbol{u}_1^{j_1} \boldsymbol{u}_2^{j_2} \cdot ... \cdot \boldsymbol{u}_n^{j_n} \tag{13}$$

This implies that the convolutional representation achieves, in fact, a generalized, higher-order notion of the mix values:

$$\forall 0 \le j_1, ..., j_n \le m - 1 \, \exists k : \quad \frac{\partial^n}{\partial \boldsymbol{u}_1^{j_1} \partial \boldsymbol{u}_2^{j_2} ... \partial \boldsymbol{u}_n^{j_n}} \boldsymbol{h}^k = 1 \tag{14}$$

This notion of higher-order neighbor mixing is visualized in Figure 2. We refer the reader to Appendix A.4 for further theoretical discussions on the stability of permutation-invariant representations.

### 4.4 Practical considerations

Combining Theorem 4.2 with an MLP compressor yields the "vanilla" version of SSMA: it first applies the local affine map, then computes 2D circular convolution across the neighbor axis and finally compresses the result back using MLP as a universal compressor. The circular convolution is implemented by applying FFT, performing product aggregation along the neighbor axis and then transforming the result back using IFFT. As "scatter_mul" is not implemented for complex numbers in standard libraries, we convert complex values to their polar representation in which multiplication is equivalent to multiplying the magnitudes and summing up the arguments. The "vanilla" version of SSMA is presented in Figure 3.

We now suggest a few practical adjustments to the "vanilla" version of SSMA:

**Normalizing the circular convolution.** As SSMA performs a product over the neighbors' axis, the optimization process of the vanilla SSMA might get unstable. To address this instability, we normalize the element-wise magnitudes of the product by taking their geometric means.

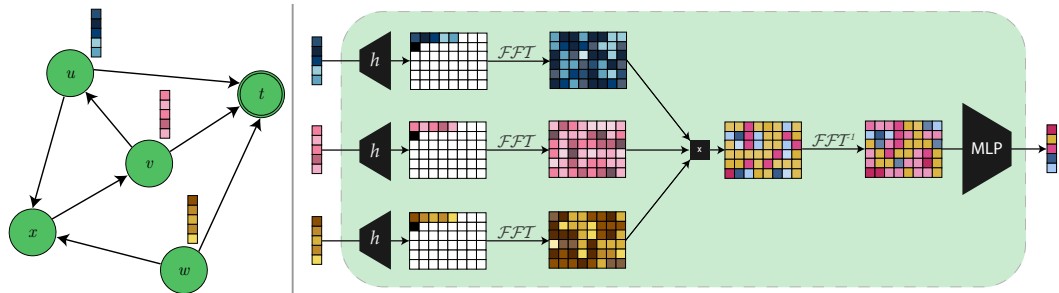

Figure 3: Visualization of the Sequential Signal Mixing Aggregation. Left: demonstration of the aggregation stage in an off-the-shelf MPGNN layer. The goal is to create a compressed view of $t$'s incoming neighbors. Right: our proposed aggregation. We convert the neighbor features into two-dimensional discrete signals. We then apply 2D circular convolution by applying 2D FFT, performing pointwise multiplication and transforming back using IFFT. Finally, we compress the result back into a $d$-dimensional vector using a multi-layer perceptron as a universal compressor.

**Low-rank compressor.** Since the number of parameters in the MLP compressor rapidly increases with the representation dimension $m$, we opted for a single linear layer as our compressor. To accommodate a higher number of neighbor slots and allow for a larger hidden dimension, we reduced the number of parameters in the linear layer by splitting it into two consecutive linear layers that squeezes the representation to low dimension and than expands it back. This effectively performs a low-rank factorization of the weight matrix of the original single linear layer.

**Neighbor selection methods.** The representation size of the vanilla SSMA $m = \mathcal{O}(n^2 d)$ may become prohibitively high in dense neighborhoods (e.g in transductive settings). To address this issue, we employ two neighbor selection techniques that reduce the original neighborhood to a new set of $\kappa$ neighbors. The first technique simply draws at most $\kappa$ random neighbors without replacement. The second technique draws inspiration from Graph Attention Networks (GAT) and attention slots [29, 42] and map the neighbors into $\kappa$ attention slots. The attention coefficient for the edge $e : j \rightarrow i$ for the $k$-th slot is expressed as follows:

$$e_k(\boldsymbol{h}_i, \boldsymbol{h}_j) = \text{LeakyReLU}(\boldsymbol{a}_k^T \boldsymbol{h}_i + \boldsymbol{b}_k^T \boldsymbol{h}_j) \tag{15}$$

$$\alpha_{ij}^{(k)} = \text{softmax}_j e_k(\boldsymbol{h}_i, \boldsymbol{h}_j) = \frac{\exp(e_k(\boldsymbol{h}_i, \boldsymbol{h}_j))}{\sum_{j' \in \mathcal{N}_{in}(i)} \exp(e_k(\boldsymbol{h}_i, \boldsymbol{h}_{j'}))} \tag{16}$$

Where $\boldsymbol{a}_k, \boldsymbol{b}_k \in \mathbb{R}^d$ are per-slot learnable weight vectors. Thereafter, the $k$-th slot for the $i$-th node is produced by considering the weighted average of the incoming neighbors:

$$s_i^{(k)} = \sum_{j \in \mathcal{N}_{in}(i)} \alpha_{ij}^{(k)} \boldsymbol{h}_j \tag{17}$$

## 5 Experiments

### 5.1 Synthetic task

To empirically demonstrate the success of SSMA in managing tasks characterized by high neighbor mixing (opposed to sum-based aggregators), we introduce a synthetic regression task we name SUMOFGRAM. In this task, we sample random neighbor features and then generate the labels by considering the sum of the Gramian matrix corresponding to the neighbor features.

In some sense, the SUMOFGRAM task is the "simplest" task that involves neighbor mixing:

$$\forall i \neq j : \text{mix}_{i,j} = \left\| \frac{\partial^2}{\partial x_i \partial x_j} \text{SUMOFGRAM}(x_1, ..., x_n) \right\|_2 = \|\mathbb{I}_{d \times d}\|_2 = 1 \tag{18}$$

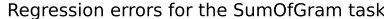

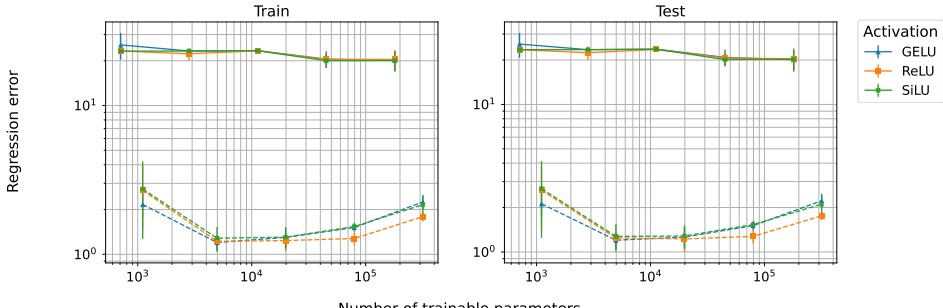

Figure 4: SUMOFGRAM train and test regression $L_1$ errors for different activation functions. The sum aggregator (not dashed) performs poorly and fails to scale with the capacity of the aggregation module, even when used in conjunction with analytic activations. On the contrary, SSMA (dashed) consistently achieves low regression errors and scales well with the number of learnable parameters.

We train both the sum aggregator and our proposed Sequential Signal Mixing Aggregation until convergence with varying representation sizes $m$ on the SUMOFGRAM task.

As may be observed in Figure 4, sum aggregators fail at this task, even when used in conjunction with analytic activations, which as claimed previously [1], is sufficient to achieve separation. This shows that sole separation is insufficient for performing arbitrary downstream tasks. On the contrary, SSMA has low regression errors, consistently along different activation functions.

## 5.2 Benchmarking SSMA

**Experimental Setup.** We test the effectiveness of SSMA by incorporating it into popular MPGNN architectures. We evaluate both original and augmented architectures across a wide range of benchmarks. These benchmarks cover learning in both the transductive and inductive settings, node and graph-level prediction tasks, regression and classification problems, feature-oriented as well as purely topological data and tasks that involve challenging neighborhood configurations including dense neighborhoods and distant neighbor dependencies. As SSMA introduces learnable parameters, we ensure a fair comparison by maintaining an equal total parameter count to that of the original architectures in each experimental setting, adjusting the architecture's hidden dimension to adhere to the budget constraints. For a detailed discussion on the parameter budget in each experiment, please refer to Appendix C.4. Given the budget for each experiment, we conduct a hyperparameter search (HPS) on SSMA parameters to find the best configuration. We further perform ablation studies to closely examine the effect of each hyperparameter, as detailed in Appendix E.

**Results.** We observe substantial performance gains across all tested combinations of benchmarks and MPGNN architectures. Notably, the most significant relative improvements were observed on the IMDB-B benchmark, which lacks node and edge features. This phenomenon is likely attributed to SSMA's neighbor mixing capabilities, enabling it to learn joint topological relationships among neighbors. The Improvements observed on the LRGB [16] datasets indicate that SSMA better extracts relevant neighborhood information to be propagated to distant parts of the graph. Additionally, the experiments on the OGBN networks (OGBN-Arxiv and OGBN-Products) [21] confirm that SSMA is robust to dense neighborhoods and highlight the efficiency of its attentional neighbor selection mechanism. Another noteworthy observation is that SSMA utilizes the hidden dimension more effectively. Since we use the same parameter budget, experiments with SSMA employ a lower hidden dimensionality than those using 'sum' aggregation. This is because SSMA allocates learnable parameters, whereas 'sum' aggregation does not. Despite a smaller hidden dimension, SSMA outperforms 'sum' aggregation, indicating its efficiency in retaining relevant information for downstream tasks. Benchmarks for more common aggregation functions is in Appendix F

Table 1: Results for TU datasets [32] & ZINC [19] using **sum** aggregation as a baseline. We report the TU datasets' accuracy mean and STD of a 10-fold cross-validation run. For the ZINC dataset, we report mean MAE and STD on the test set according to 5 distinct runs. [†] indicates reproduced results while [*] indicates the reported results from the relevant paper.

| Module | ENZYMES ↑ | PTC-MR ↑ | MUTAG ↑ | PROTEINS ↑ | IMDB-B ↑ | ZINC ↓ |
|---|---|---|---|---|---|---|
| GCN[†] [24] | 51.0±10.63 | 59.85±4.04 | 84.23±9.86 | 75.39±4.53 | 68.80±3.49 | 0.347±0.01 |
| GCN + SSMA | **54.83±7.55** | **62.29±9.33** | **89.79±6.71** | **76.28±3.19** | **75.2±2.9** | **0.280±0.02** |
| GAT[†] [43] | 50.67±4.92 | 65.53±8.41 | 75.51±11.72 | 73.32±3.08 | 51.0±6.07 | 0.386±0.025 |
| GAT + SSMA | **56.67±3.72** | **66.41±5.69** | **89.19±4.58** | **80.18±0.1** | 74.5±4.14 | **0.223±0.028** |
| GATv2[†] [6] | 44.83±5.96 | 56.47±7.57 | 77.26±13.15 | 73.04±3.35 | 47.0±5.27 | 0.396±0.006 |
| GATv2 + SSMA | **52.50±8.43** | **61.64±6.80** | **88.80±11.80** | **75.28±4.80** | **72.8±4.92** | **0.235±0.003** |
| GIN[†] [48] | 49.50±4.58 | 60.46±9.10 | 86.45±8.17 | 73.30±5.11 | 71.3±3.97 | 0.252±0.007 |
| GIN + SSMA | **51.69±8.04** | **61.28±9.23** | **90.51±6.97** | **75.19±4.73** | **74.1±5.02** | **0.222±0.003** |
| GraphGPS[†] [37] | 48.33±6.71 | 61.41±6.91 | 79.91±10.23 | 73.76±6.05 | 69.6±5.54 | 0.251±0.012 |
| GraphGPS + SSMA | **49.17±3.15** | **63.02±4.93** | **86.07±7.95** | **75.56±4.24** | **71.1±4.79** | **0.22±0.005** |
| PNA[†] [12] | 52.50±4.60 | 58.41±6.66 | 84.19±9.44 | 74.86±4.57 | 71.9±4.46 | 0.192±0.001 |
| PNA + SSMA | **52.92±7.34** | **62.14±5.54** | **88.29±8.46** | **75.68±5.91** | **74.1±4.23** | **0.172±0.001** |
| ESAN[*] [3] | - | 69.2±6.5 | 91.1±7.0 | 75.9±4.3 | 77.1±3.0 | 0.102±0.003 |
| ESAN + SSMA | - | **77.89±5.62** | **96.32±3.37** | **80.69±4.1** | **80.6±2.15** | **0.096±0.002** |
| Improvement (%) | 7.2 | 5.3 | 8.9 | 3.7 | 17.7 | 20.36 |

Table 2: Test performance on the OGB [21] & LRGB [16] benchmarks using **sum** aggregation as a baseline. [†] indicates reproduced results while [*] indicates the reported results from the relevant paper.

| Module | LRGB | | OGB-N | | OGB-G | |
|---|---|---|---|---|---|---|
| | Peptides-f | Peptides-s | Arxiv | Products | molhiv | molpcba |
| | AP ↑ | MAE ↓ | Accuracy ↑ | Accuracy ↑ | AUROC ↑ | AP ↑ |
| GCN[†] [24] | 61.1±1.04 | 0.28±0.01 | 65.6±0.55 | 63.8±3.45 | 0.77±0.01 | 0.21±0.01 |
| GCN + SSMA | **63.3±1.42** | **0.26±0.02** | **66.3±0.48** | **72.3±3.94** | **0.79±0.01** | **0.23±0.01** |
| GAT[†] [43] | 63.4±0.68 | 0.27±0.01 | 62.1±0.64 | 60.6±7.65 | 0.75±0.02 | 0.21±0.01 |
| GAT + SSMA | **63.6±0.47** | **0.26±0.01** | **66.6±0.78** | **67.3±5.81** | **0.79±0.01** | **0.22±0.01** |
| GATv2[†] [6] | 63.1±1.34 | 0.27±0.01 | 62.8±0.85 | 56.7±8.25 | 0.75±0.01 | 0.18±0.01 |
| GATv2 + SSMA | **63.7±1.13** | **0.26±0.01** | **64.7±0.62** | **66.4±3.70** | **0.79±0.01** | **0.22±0.01** |
| GIN[†] [48] | 60.4±0.96 | 0.27±0.01 | 54.1±0.87 | 54.8±5.53 | 0.75±0.01 | 0.21±0.01 |
| GIN + SSMA | **62.5±1.37** | **0.26±0.02** | **66.4±1.52** | **67.0±5.79** | **0.78±0.01** | **0.22±0.01** |
| GraphGPS[†] [37] | 58.81±1.22 | 0.28±0.01 | 63.87±0.68 | 48.89±7.47 | 0.76±0.02 | 0.19±0.01 |
| GraphGPS + SSMA | **60.34±1.49** | 0.27±0.01 | **66.71±0.73** | **67.62±5.46** | **0.78±0.01** | **0.22±0.01** |
| PNA[†] [12] | 57.0±1.17 | 0.28±0.01 | 59.1±0.60 | 45.6±16.52 | 0.75±0.05 | 0.17±0.01 |
| PNA + SSMA | **61.1±1.75** | 0.27±0.03 | **66.3±0.81** | **63.9±3.72** | **0.78±0.02** | **0.21±0.01** |
| ESAN[*] [3] | - | - | - | - | 0.76±0.01 | - |
| ESAN + SSMA | - | - | - | - | **0.83±0.01** | - |
| Improvement (%) | 3.02 | 4.21 | 8.9 | 23.86 | 4.62 | 13.4 |

# 6 Conclusion and discussion

In this work, we re-examined the field of aggregation functions in MPGNNs and introduced the Sequential Signal Mixing Aggregation, a new plug-and-play aggregation method with a solid theoretical foundation. We demonstrated its effectiveness across various datasets and message-passing architectures with different parameter budgets. Each component of our method was systematically examined and its contribution to performance is verified. We hope the observed performance gains will motivate further research into harnessing our aggregation for specific applications and developing more advanced aggregation functions for MPGNNs. Future research could address some limitations of our method, such as the representation size scaling quadratically with the number of neighbors and the need for explicit normalization due to the instability of the convolution operation.

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

# A  Proofs and extended theory discussion

## A.1  Proof of Proposition 3.2

*Proof.* Let us consider a general sum-based aggregator: $F(x_1, ..., x_n) = \rho(\sum_{k=1}^{n} \phi(x_k))$.

Then, we have that for $i \neq j$ and for the $\ell$-th output:

$$\text{mix}_{i,j}^{(\ell)} = \tag{19}$$

$$\left\| \frac{\partial^2}{\partial x_i \partial x_j} \rho^{(\ell)}(\sum_{k=1}^{n} \phi(x_k)) \right\|_2 = \tag{20}$$

$$\left\| \frac{\partial}{\partial x_j} \left\{ \frac{\partial}{\partial x_i} \rho^{(\ell)}(\sum_{k=1}^{n} \phi(x_k)) \right\} \right\|_2 = \tag{21}$$

$$\left\| \frac{\partial}{\partial x_j} \left\{ \nabla \rho^{(\ell)}(\sum_{k=1}^{n} \phi(x_k)) \cdot J_\phi(x_i) \right\} \right\|_2 = \tag{22}$$

$$\left\| \frac{\partial}{\partial x_j} \left\{ \nabla \rho^{(\ell)}(\sum_{k=1}^{n} \phi(x_k)) \right\} \cdot J_\phi(x_i) \right\|_2 = \tag{23}$$

$$\left\| J_\phi(x_j)^T \cdot H_{\rho^{(\ell)}}(\sum_{k=1}^{n} \phi(x_k)) \cdot J_\phi(x_i) \right\|_2 \leq \tag{24}$$

$$\|J_\phi(x_i)\|_2 \cdot \left\| H_{\rho^{(\ell)}}(\sum_{k=1}^{n} \phi(x_k)) \right\|_2 \cdot \|J_\phi(x_j)\|_2 \tag{25}$$

$\square$

Typically, the local operator $\phi : \mathbb{R}^d \to \mathbb{R}^m$ is of the form $\phi(x) = \sigma(Ax + b)$ where $\sigma$ is an activation function applied element-wise and $A \in \mathbb{R}^{m \times d}, b \in \mathbb{R}^m$ are learnable parameters.

Usually $|\sigma'(z)|$ is bounded by some small constant $c_1$.

Therefore, we have:

$$\|J_\phi(x)\|_2 = \|\text{diag}(\sigma'(Ax + b)) \cdot A\| \leq c_1 \cdot \|A\|_2 \tag{26}$$

Moreover, the global pooling operator $\rho : \mathbb{R}^m \to \mathbb{R}^d$ is an MLP of the form:

$$\rho(z) = W_2 \cdot \sigma(W_1 z + \beta_1) + \beta_2 \tag{27}$$

where $W_1 \in \mathbb{R}^{m \times m}, \beta_1 \in \mathbb{R}^m, W_2 \in \mathbb{R}^{d \times m}, \beta_2 \in \mathbb{R}^d$.

Therefore:

$$\frac{d}{dz} \rho^{(\ell)}(z) = W_2[\ell, :]^T \cdot \text{diag}(\sigma'(W_1 z + \beta_1)) \cdot W_1 \tag{28}$$

Let us denote:

$$u^{(\ell)}(z) := W_2[\ell, :]^T \cdot \text{diag}(\sigma'(W_1 z + \beta_1)) = (W_2[\ell, k] \cdot \sigma'([W_1 z + \beta_1]_k))_{k=1}^{m} \tag{29}$$

Then,

$$\frac{d^2}{dz^2} \rho^{(\ell)}(z) = \frac{d}{dz} u^{(\ell)}(z) \cdot W_1 \tag{30}$$

We compute $\frac{d}{dz} u^{(\ell)}(z)$ for each output dimension $1 \leq k \leq m$ separately:

$$\frac{d}{dz}\left\{u^{(\ell)}(z)[k]\right\} = \frac{d}{dz}\left\{W_2[\ell,k]\cdot\sigma'(W_1[k,:]z+\beta_1[k])\right\} = \tag{31}$$

$$W_2[\ell,k]\cdot\sigma''(W_1[k,:]z+\beta_1[k])\cdot W_1[k,:] \tag{32}$$

Summarizing this for all dimensions $k$ yields:

$$\frac{d}{dz}u^{(\ell)}(z) = \mathsf{diag}_k(W_2[\ell,k])\cdot\mathsf{diag}(\sigma''(W_1z+\beta_1))\cdot W_1 \tag{33}$$

So under the assumption $|\sigma''(z)|\leq c_2$ we get the following bound on the Hessian norm:

$$\left\|H_{\rho^{(\ell)}}(z)\right\|_2 \leq c_2\cdot\|W_2[\ell,:]\|_2\cdot\|W_1\|_2^2 \tag{34}$$

And the final bound on $\mathsf{mix}_{i,j}^{(\ell)}$ is given by:

$$\mathsf{mix}_{i,j}^{(\ell)} \leq c_2\cdot c_1^2\cdot\|W_2[\ell,:]\|_2\cdot\|W_1\|_2^2\cdot\|A\|_2^2 \in \mathcal{O}(\|\theta\|_2^2) \tag{35}$$

## A.2 Ring theory and the factorization lemma

A **ring** is an algebraic structure that generalizes the notion of a field. In particular, univariate and multivariate polynomials obey this structure. Formally, a ring $R$ is a set associated with two binary operations $+$ (addition) and $\cdot$ (multiplication) satisfying the ring axioms:

1. $R$ is an abelian group under the addition operation.

2. $R$ is a monoid under the multiplication operation:

   (a) associativity: $(a \cdot b) \cdot c = a \cdot (b \cdot c)$ for all $a, b, c \in R$.
   (b) existence of identity: there is an element $1 \in R$ such that: $1 \cdot a = a \cdot 1 = a$.

3. Distributivety: $a \cdot (b + c) = a \cdot b + a \cdot c$ and $(b + c) \cdot a = b \cdot a + c \cdot a$.

A ring is said to be **commutative** if its elements commute under multiplication: $a \cdot b = b \cdot a$ for all $a, b \in R$.

Given a commutative ring $R$, we can define its corresponding univariate **polynomial ring** denoted as $R[X]$ by considering a set of formal expressions $\sum_{i=0}^{n} \alpha_i X^i$ where $n$ is a non-negative integer and $\alpha_i \in R$. We consider $X$ a formal variable and define addition and multiplication according to the ordinary rules for manipulating algebraic expressions. Each polynomial $p \in R[X]$ has a **degree** defined as $\max_i \alpha_i \neq 0$. For each $r \in R$, we can define an **evaluation map** $T_r$ which takes some polynomial as an input and returns an element in the underlying ring by substituting $X = r$, namely: $T_r\left(\sum_{i=0}^{n} \alpha_i X^i\right) = \sum_{i=0}^{n} \alpha_i r^i$.

**Multivariate polynomials** can be defined similarly by considering multiple formal variables. Alternatively, note that since the polynomial ring of some commutative ring $R$ is also a commutative ring by itself, we can equivalently define multivariate polynomials by considering the ring of polynomials above $R[X]$, namely: $(R[X])[Y] \cong R[X, Y]$.

An **integral domain** is a nonzero commutative ring in which the product of any two nonzero elements is nonzero. In particular, any field is an integral domain, and any polynomial ring is an integral domain, given that its underlying ring is itself an integral ring. Am immediate conclusion is that $\mathbb{R}[x]$ and $\mathbb{R}[x_1, ..., x_n]$ are integral domains.

An **irreducible element** of an integral domain is a non-zero element that is not invertible and is not the product of two non-invertible elements. For instance, for every commutative ring $R$, every polynomial of the form $x - r$ where $r \in R$ is an irreducible element of $R[x]$.

A **unique factorization domain** (UFD) is an integral domain $R$ in which every non-zero element $r$ of $R$ can be written as a product (an empty product if $x$ is invertible) of irreducible elements $p_i$ of $R$ and an invertible element $u$: $r = u \cdot \prod_{i=1}^{n} p_i$. This representation ought to be unique up to multiplication with invertible elements. The key fact underlying our construction is the fact that a polynomial ring of a UFD is by itself a UFD (known as **Gauss's lemma**). In conjunction with the fact that any field $\mathbb{F}$ is a UFD, we get that $\mathbb{R}[x_1, ..., x_n]$ is a UFD for any amount of variables.

**Lemma A.1.** *Let $p(t, z) \in \mathbb{R}[t, z]$ be some polynomial that can be factorized as:*

$$p(t, z) = \prod_{i=1}^{n} (t - u_i(z)) \tag{36}$$

*where $u_i(z)$ is some polynomial of $z$.*

*Then, such factorization is unique to the order of the terms $(t - u_i(z))$.*

*Proof.* Gauss's lemma implies that $\mathbb{R}[t, z] \cong (\mathbb{R}[z])[t]$ is a unique factorization domain. Since every polynomial of the form $t - r(z)$ is an irreducible element in $(\mathbb{R}[z])[t]$, it follows that a factorization of the form $p(t, z) = \prod_i (t - q_i(z))$ is unique up to permutation of the terms. $\square$

**Corollary A.2.** *The coefficients of the polynomial $p_{\overline{\mathbf{X}}}(t, z)$ defined in Section 4.2 form an ensemble of separating invariants.*

## A.3 Proof of Theorem 4.2

*Proof.* Let $p_{\overline{\mathbf{X}}}(t, z)$ be the polynomial in the construction. Corollary A.2 implies that its coefficients form an ensemble of separating invariants. Consequently, we repeat the steps in Section 4.1 to get an analogous result to Theorem 4.1, arriving at the desired form.

We represent $p_{\overline{\mathbf{X}}}(t, z)$ by evaluating its value on a grid of points $(u_s, v_b)$ where $0 \le s \le n, 0 \le b \le \tau$. Specifically, we can choose $u_s := e^{-\frac{2\pi i}{n+1} s}$ and $v_b := e^{-\frac{2\pi i}{\tau+1} b}$ and to get the two-dimensional DFT of the polynomial coefficients:

$$p_{\overline{\mathbf{X}}}(u_s, v_b) = \sum_{k=0}^{n} \sum_{\ell=0}^{\tau} e_{k\ell}(\mathbf{X}) \cdot (u_s)^k (v_b)^\ell = \sum_{k,\ell} e_{k\ell}(\mathbf{X}) \cdot e^{-\frac{2\pi i s}{n+1} k} \cdot e^{-\frac{2\pi i b}{\tau+1} \ell} \tag{37}$$

Again, by setting $\mathbf{\Phi}(\mathbf{X}_i)$ to be the coefficients matrix of $p_i(t, z) = t - q_{\mathbf{X}_i}(z)$:

$$\mathbf{\Phi}(\mathbf{X}_i) = \begin{bmatrix} -\mathbf{X}_{i1} & -\mathbf{X}_{i2} & \cdots & -\mathbf{X}_{id} & \cdots & 0 \\ 1 & 0 & 0 & 0 & \cdots & 0 \\ \vdots & \vdots & \vdots & \vdots & \ddots & 0 \\ 0 & 0 & 0 & 0 & \cdots & 0 \end{bmatrix} \tag{38}$$

we get that:

$$e_{k\ell}(\mathbf{X}) = \mathcal{F}_{2D}^{-1} \left\{ \bigodot_{i=1}^{n} \mathcal{F}_{2D} \left\{ \mathbf{\Phi}(\mathbf{X}_i) \right\} \right\} \tag{39}$$

Where the multiplication is elementwise. According to the 2D circular convolution theorem, this exactly amounts to convolving the vector coefficients of $p_i(t, z)$s:

$$e_{k\ell}(\mathbf{X}) = \circledast_{i=1}^{n} \mathbf{\Phi}(\mathbf{X}_i) \tag{40}$$

$\square$

## A.4 On the stability of permutation-invariant representations

Ideally, we would like our representation to be numerically stable. That is, to require that if two distinct multisets are close to each other, then their representations should also be close to each other, and vice versa. This was formalized previously [1] using the notions of bi-Lipschitzness and Wasserstein distance as we now recapitulate.

Let $\overline{\mathbf{X}}, \overline{\mathbf{Y}}$ be two multisets of $d$ dimensional vectors with $|\overline{\mathbf{X}}| = |\overline{\mathbf{Y}}| = n$.

**Wasserstein distance.** we measure the distance between equally sized multisets $\subseteq \mathbb{R}^d$ using the notion of Wasserstein distance:

$$\mathcal{W}_p(\overline{\mathbf{X}}, \overline{\mathbf{Y}}) := \min_{\pi \in S_n} \left(\frac{1}{n} \sum_{i=1}^n \left\|\mathbf{X}_i - \mathbf{Y}_{\pi(i)}\right\|^p\right)^{1/p} \tag{41}$$

Where $\|.\|$ is the $L_1$ norm over $\mathbb{R}^d$.

We are interested in the bi-Lipschitzness property - whether there exist constants $c, C > 0$ such that:

$$c \cdot \mathcal{W}_p(\overline{\mathbf{X}}, \overline{\mathbf{Y}}) \leq \left\|\hat{f}(\mathbf{X}) - \hat{f}(\mathbf{Y})\right\| \leq C \cdot \mathcal{W}_p(\overline{\mathbf{X}}, \overline{\mathbf{Y}}) \tag{42}$$

Unfortunately, it turns out that this notion of stability is unattainable by any differentiable permutation-invariant representation. We prove this by generalizing such results for sum-based aggregators [1].

**Proposition A.3.** *Let $\hat{f}$ be some differentiable multiset representation. Then, there exist $n, d$ such that for every $\epsilon > 0$ there exist $\overline{\mathbf{X}}_\epsilon, \overline{\mathbf{Y}}_\epsilon \subseteq \mathbb{R}^d$ two multisets of size $n$ such that:*

$$\left\|\hat{f}(\mathbf{X}_\epsilon) - \hat{f}(\mathbf{Y}_\epsilon)\right\| \leq \epsilon \cdot \mathcal{W}_p(\overline{\mathbf{X}}_\epsilon, \overline{\mathbf{Y}}_\epsilon) \tag{43}$$

An independent proof for general invariant embeddings is given in [8].

*Proof.* Let $\hat{f}$ be some permutation-invariant representation. We use the same construction as in [1] and generalize the proof to arbitrary symmetric and differential representations.

We choose $n = 2$ and some arbitrary $d \in \mathbb{N}$.

Let $\mathbf{x_0}, \mathbf{d} \in \mathbb{R}^d$ where $\mathbf{d}$ has a unit norm, and consider $S_h = \{\{\mathbf{x_0} + h\mathbf{d}, \mathbf{x_0} - h\mathbf{d}\}\}$.

We note that the Wasserstein distance $\mathcal{W}_p(S_h, S_0)$ is $h$. Thus, it is sufficient to show that:

$$\lim_{h \to 0} \frac{\left\|\hat{f}(S_h) - \hat{f}(S_0)\right\|}{h} = 0 \tag{44}$$

Since $\hat{f}_k : \mathbb{R}^{2d} \to \mathbb{R}$ is invariant we have that for every $\mathbf{u}, \mathbf{v} \in \mathbb{R}^d$: $\hat{f}_k(\mathbf{u}, \mathbf{v}) = \hat{f}_k(\mathbf{v}, \mathbf{u})$.

The differentiability of $\hat{f}_k$ at $(\mathbf{u}, \mathbf{v})$ implies that for every $1 \leq i \leq d$:

$$\partial_i \hat{f}_k(\mathbf{u}, \mathbf{v}) = \lim_{h \to 0} \frac{\hat{f}_k(u_1, ..., u_i + h, ..., u_d, \mathbf{v}) - \hat{f}_k(\mathbf{u}, \mathbf{v})}{h} = \tag{45}$$

$$\lim_{h \to 0} \frac{\hat{f}_k(\mathbf{v}, u_1, ..., u_i + h, ..., u_d) - \hat{f}_k(\mathbf{v}, \mathbf{u})}{h} = \partial_{d+i} \hat{f}_k(\mathbf{v}, \mathbf{u}) \tag{46}$$

Particularly, this implies that $\partial_i \hat{f}_k(\mathbf{x_0}, \mathbf{x_0}) = \partial_{d+i} \hat{f}_k(\mathbf{x_0}, \mathbf{x_0})$.

Using the differentiability of $\hat{f}_k$ at $(\mathbf{x_0}, \mathbf{x_0})$ we can write:

$$\hat{f}_k(\mathbf{x_0} + \delta_1, \mathbf{x_0} + \delta_2) = \tag{47}$$

$$\hat{f}_k(\mathbf{x_0}, \mathbf{x_0}) + \sum_{i=1}^d \partial_i \hat{f}_k(\mathbf{x_0}, \mathbf{x_0}) \cdot \delta_{1i} + \sum_{i=1}^d \partial_{d+i} \hat{f}_k(\mathbf{x_0}, \mathbf{x_0}) \cdot \delta_{2i} + o_k(\|(\delta_1, \delta_2)\|) \tag{48}$$

Plugging in $\delta_1 = h\mathbf{d}$ and $\delta_2 = -h\mathbf{d}$ we get:

$$\hat{f}_k(\mathbf{x_0} + h\mathbf{d}, \mathbf{x_0} - h\mathbf{d}) - \hat{f}_k(\mathbf{x_0}, \mathbf{x_0}) = \tag{49}$$

$$h\sum_{i=1}^{d} \partial_i \hat{f}_k(\mathbf{x_0}, \mathbf{x_0}) \cdot d_i - h\sum_{i=1}^{d} \partial_{d+i} \hat{f}_k(\mathbf{x_0}, \mathbf{x_0}) \cdot d_i + o_k(h) = o_k(h) \tag{50}$$

All in all, we have:

$$\lim_{h\to 0} \frac{\left\| \hat{f}(S_h) - \hat{f}(S_0) \right\|}{h} = \lim_{h\to 0} \frac{\sum_{k=1}^{m} |\hat{f}_k(\mathbf{x_0} + h\mathbf{d}, \mathbf{x_0} - h\mathbf{d}) - \hat{f}_k(\mathbf{x_0}, \mathbf{x_0})|}{h} = \tag{51}$$

$$\lim_{h\to 0} \frac{\sum_{k=1}^{m} |o_k(h)|}{h} = \sum_{k=1}^{m} \lim_{h\to 0} \frac{|o_k(h)|}{h} = 0 \tag{52}$$

Meaning that for every $\epsilon > 0$ there exists sufficiently small $h$ such that:

$$\left\| \hat{f}(S_h) - \hat{f}(S_0) \right\| \le \epsilon \cdot h = \epsilon \cdot \mathcal{W}_p(S_h, S_0) \tag{53}$$

$\square$

# B  Complexity analysis

## B.1  Theoretical analysis

In this section, we provide a theoretical complexity analysis of SSMA for both its "vanilla" and refined versions. We consider the total cost of the aggregation stage in a single MPGNN layer within a graph $G = (V, E)$. We let $d$ be the hidden dimension and $m = m_1 \cdot m_2$ be the total representation size. We let $\kappa$ be the number of slots in the refined version of SSMA. We first analyze the vanilla version of SSMA step-by-step:

1. **Local affine layer:** locally transforms each node by an affine transformation $\mathcal{O}(|V| \cdot m \cdot d)$.
2. **Local FFT:** the FFT is computed per node yielding a cost of $\mathcal{O}(|V| \cdot m \log(m))$.
3. **Product aggregation:** The complex variant of `scatter_mul` aggregation - $\mathcal{O}(|E| \cdot m)$.
4. **Local IFFT:** Same as stage 2.
5. **MLP compressor:** We used linear layer as a compressor to the original dimension - $\mathcal{O}(|V| \cdot m \cdot d)$.

All in all we get: $\mathcal{O}(m(d + \log(m))|V| + m|E|)$ compared to the standard $\mathcal{O}(md|V| + m|E|)$ which is a negligible slowdown.

In the refined version of SSMA, we consider the modifications we applied:

- **Neighbor selection:** If the random neighbor selection is used, then the cost is simply $\mathcal{O}(m \cdot \kappa \cdot |V| + |E|)$ for this stage. Alternatively, if the soft-attentional neighbor selection is picked, the attention weights in Equation (16) may be computed per edge and attention slot in a total running time of $\mathcal{O}(m \cdot \kappa \cdot |V| + \kappa \cdot |E|)$ and the contents of the slots may be implemented using sum aggregation in $\mathcal{O}(m \cdot \kappa \cdot |E|)$. We get a total of $\mathcal{O}(m \cdot \kappa \cdot |V| + |E|)$ for the random neighbor selection or $\mathcal{O}(m \cdot \kappa \cdot (|V| + |E|))$ for the soft-attentional neighbor selection for this stage.

- **Other modifications:** the normalization method does not affect the complexity of the aggregation. While the complexity of the MLP compressor is slightly reduced, the asymptotic complexity remains the same due to the local affine layer bottleneck.

- **Application of SSMA on the new neighborhood**: We consider the complexity obtained in the analysis of the vanilla SSMA and replace the number of edges, $|E|$, with $\kappa \cdot |V|$. The complexity of this stage then becomes: $\mathcal{O}(m(d + \log(m))|V| + m \cdot \kappa \cdot |V|) = \mathcal{O}(m(d + \log(m) + \kappa)|V|)$.

All in all, we get $\mathcal{O}(m(d + \log(m) + \kappa)|V| + |E|)$ for the random neighbor selection refined SSMA or $\mathcal{O}(m(d + \log(m) + \kappa)|V| + m\kappa|E|)$ for the soft neighbor selection refined version of SSMA.

## B.2  Runtime measurement

To further show that SSMA is scalable we computed the runtime of it compared to other common networks, we measured the runtime of the layer with our aggregation with and without attention, for each configuration we show the downstream task results. As can be seen in Table 3 SSMA runtime is comparable to other methods while achieving higher downstream task performance.

Table 3: Comparison of the training and inference times in (ms) of MPGNNs+SSMA against PNA and GraphGPS. rSSMA indicates random neighbor selection while aSSMA indicates attentional neighbor selection.

| Method | Train (ms) | Inference (ms) | Arxiv Acc. (%) | Proteins Acc. (%) |
|---|---|---|---|---|
| GIN + rSSMA | 2.768 | 0.988 | 64.217 | 73.921 |
| GCN + rSSMA | 5.286 | 1.322 | 64.834 | 74.308 |
| GAT + rSSMA | 5.996 | 1.453 | 64.902 | 75.197 |
| PNA | 6.068 | 1.218 | 59.1 | 74.86 |
| GPS | 7.178 | 1.362 | 63.87 | 73.76 |
| GIN + aSSMA | 8.072 | 1.504 | 65.835 | 72.94 |
| GCN + aSSMA | 8.347 | 1.666 | 65.368 | 73.132 |
| GAT + aSSMA | 9.236 | 1.888 | 64.108 | 72.677 |

## C  Extended experimental setup

### C.1  Datasets

**TU Datasets [32].**    We conducted experiments on five widely used datasets, including four bioinformatics datasets (ENZYMES, PTC-MR, MUTAG, and PROTEINS) and one movie collaboration dataset that relies solely on topological data (IMDB-Binary). While these datasets are all relatively small, as detailed in Table 4, they span a diverse range of domains.

The ENZYMES dataset consists of 600 enzyme graphs, where the task is to classify each enzyme into one of six EC top-level classes. Nodes represent secondary structure elements (SSEs) and are annotated by type. An edge connects two nodes if they are neighbors along the amino acid sequence or among the three nearest spatial neighbors, with edge features indicating their type (structural or sequential).

The PTC-MR dataset contains 344 compounds labeled according to their carcinogenicity in male rats. The MUTAG dataset consists of 188 chemical compounds, divided into two classes based on their mutagenic effect on bacteria. The PROTEINS dataset includes 1,113 protein molecules, with the task being a multiclass protein function prediction. For all of the above datasets, nodes represent atoms, while edges mean that two atoms are connected.

As these datasets lack official train/validation/test splits, we employ random 10-fold cross-validation for evaluation. The prediction task for all these datasets is multiclass classification, and we use accuracy as the performance metric.

The IMDB-Binary dataset is a movie collaboration dataset comprising 1,000 graphs. In each graph, nodes represent actors/actresses, with edges indicating co-appearance in movies. Collaboration graphs for the Action and Romance genres were generated as ego networks for each actor/actress, and each ego network was labeled with the corresponding genre. The task is to identify the genre of an ego-network graph.

**ZINC dataset [19, 22]**    . We benchmark on the ZINC dataset. Specifically, the subset of 12K graphs as defined in [15], from the full 250K ZINC dataset. The prediction task is to regress a molecular property known as constrained solubility, which is calculated as "logP - SA - cycle" (octanol-water partition coefficients, logP, penalized by the synthetic accessibility score, SA, and the number of long cycles, cycle). In this dataset, node features represent the types of heavy atoms, and edge features represent the bonds between them. The ZINC dataset is widely used for research in molecular graph generation. We used the dataset versions available via PYG without any further preprocessing.

**OGB - Graph property prediction [21].**    We conduct experiments on the commonly used ogbg-molhiv and ogbg-molpcba molecule datasets to assess SSMA on graph-level prediction tasks. In these datasets, each graph represents a molecule, with nodes corresponding to atoms and edges to chemical bonds. The node features are 9-dimensional, encompassing atomic number, chirality, and other atomic attributes such as formal charge and ring membership. We refer the reader to code for further details. The prediction task involves accurately predicting molecular properties, represented as

binary labels, such as whether a molecule inhibits HIV virus replication. The ogbg-molpcba dataset includes multiple tasks, some of which may contain 'nan' values indicating that the corresponding label is not assigned to the molecule. These datasets differ in size: ogbg-molhiv is smaller, while ogbg-molpcba is medium-sized. The evaluation metrics used are ROC-AUC for ogbg-molhiv and Average Precision (AP) for ogbg-molpcba. We used the official train/validation/test splits provided by the OGB team.

**OGB - Node property prediction [21].** We benchmark on two dense node-level citation graph datasets from OGB-N: ogbn-products and ogbn-arxiv. The login-products dataset is an undirected and unweighted graph representing an Amazon product co-purchasing network. Nodes represent products sold on Amazon, and edges indicate co-purchased products. Node features are generated by extracting bag-of-words features from product descriptions, followed by Principal Component Analysis for dimensionality reduction. The prediction task is multiclass classification to predict a product's category.

The ogbn-arxiv dataset is a smaller directed graph representing the citation network among Computer Science papers indexed by the Microsoft Academic Graph (MAG). Each node represents an arXiv paper, and the directed edges indicate citations between papers. Node embeddings are created by averaging the embeddings of words in the paper titles and abstracts, computed using the skip-gram model over the MAG corpus. The task is to predict the 40 subject areas of the arXiv CS papers.

We used the graph obtained from the OGB python package without any preprocessing.

**Long Range Graph Benchmark (LRGB) [16].** We benchmark on two graph-level datasets from the Long-Range Graph Benchmark (LRGB): peptides-func and peptides-struct. Each graph in these datasets represents a peptide, a short chain of amino acids shorter than proteins and abundant in nature. Each amino acid is composed of many heavy atoms, making the molecular graph of a peptide much larger than that of a small drug-like molecule. Peptide graphs have a low average node degree. Still, they have significantly larger diameters compared to other drug-like molecules, making them ideal for studying long-range dependencies in Graph Neural Networks (GNNs). Both datasets use the same set of graphs but differ in their prediction tasks. Peptides-func is a multi-label graph classification dataset based on peptide function, while Peptides-struct is a multi-label graph regression dataset based on the 3D structure of peptides. More details can be found in the LRGB GitHub repository. We used the versions of the datasets available via PYG without any further preprocessing.

**Dataset statistics.** The statistics of the datasets we used in our experiments are shown in Table 4.

Table 4: The statistics of the datasets used in our experiments

| Dataset | Avg. Nodes | Avg. Edges | # Graphs | Avg. in deg | STD in deg | Median deg |
|---|---|---|---|---|---|---|
| ogbg-molhiv | 25.51 | 54.94 | 41127 | 2.15 | 0.77 | 2 |
| ogbg-molpcba | 25.97 | 56.22 | 437929 | 2.16 | 0.71 | 2 |
| ogbn-arxiv | 169343 | 1166243 | 1 | 6.89 | 67.6 | 1 |
| ogbn-products | 2,449,029 | 123,718,280 | 1 | 50.52 | 95.91 | 26 |
| mutag | 17.93 | 39.59 | 188 | 2.21 | 0.74 | 2 |
| enzymes | 32.63 | 124.27 | 600 | 3.81 | 1.15 | 4 |
| proteins | 39.06 | 145.63 | 1113 | 3.73 | 1.15 | 4 |
| ptc-mr | 14.29 | 29.38 | 344 | 2.06 | 0.81 | 2 |
| imdb-binary | 19.77 | 193.06 | 1000 | 9.76 | 7.43 | 7 |
| zinc | 23.16 | 49.83 | 12000 | 2.15 | 0.72 | 2 |
| peptides-func | 150.94 | 307.3 | 15535 | 2.04 | 0.79 | 2 |
| peptides-struct | 150.94 | 307.3 | 15535 | 2.04 | 0.79 | 2 |

## C.2  Preprocessing

We did not modify the features of the graph topology in any of the experiments except in the following cases:

- For purely topological datasets lacking node features, we assigned the zero vector as the initial node feature.

- For the large graphs "OGBN-arxiv" and "OGBN-Products," we undirected the graphs and then clustered them following [9]. We did it so we will be able to fit them into memory.

- In the ESAN experiments [3], we followed the preprocessing steps outlined by the authors, as these were integral parts of their suggested method. Specifically, we used the configuration that yielded the best empirical results reported in the paper, "DSS-GNN (GIN) (EGO+)."

## C.3  Tailoring GNN architectures to different benchmarks

In all our experiments, we used consistent architectures with minor variations tailored to each dataset. The primary difference is in the initial layer, which adapts to the specific characteristics of the dataset's node and edge features. This initial layer can be either an embedding layer or a linear layer designed to project the input into the network's hidden space. Following the initial layer, the architecture consists of stacked message-passing layers with residual connections and batch normalization layers. For graph-prediction benchmarks, a readout function is applied. Finally, a two-layer multilayer perceptron (MLP) with ReLU activation produces the output.

### C.3.1  GraphGPS configuration

Due to the flexibility of GraphGPS and its numerous configuration options, we selected and focused on a specific setup. We utilized the Residual Gated Graph ConvNet from [5] as our convolution operator.For the attention module we used, dropout rate of 0.5 and 4 attention heads. For the experiments that uses Positional Encoding we used random walk with length 20. This configuration was taken from graph-gps repository. The full initialization details are available in our code.

## C.4  Parameter budget

We made our best effort to find the most common parameter budget for each benchmark. Specifically:

- For the ZINC [19] dataset, we employed a 100k parameter budget in order to obtain a fair comparison to previous works [12, 38, 3, 15].

- For the TUDatasets (MUTAG, ENZYMES, PROTEINS, IMDB-BINARY), where there is no consensus on the parameter budget, we used a 500,000 parameter budget. An exception was made for ESAN, for which we used a 100,000-parameter budget to allow fair comparison.

- For the OGB-N datasets, we used a 500,000 parameter budget. This is reasonable when considering the parameter counts of models on the ogbn-arxiv leaderboard and ogbn-products leaderboard.

- For OGB-G datasets, we used 500,000 parameters for ogbg-molhiv and 2,000,000 for ogbg-molpcba since its prediction task is more complex and contains 128 label prediction tasks. These are reasonable numbers as can be seen from ogbg-molhiv leaderboard & ogbg-molpcba leaderboard

- For the LRGB datasets, we also used a 2,000,000 parameter budget as those datasets require deeper GNNs because they need to embed long-range dependencies in the graph.

A summary of the parameter budget can be seen in Table 5

Table 5: Parameter budget used for each dataset. *for the ESAN, we used a 300,000 parameter budget. **for the ESAN we used a 100,000 parameter budget.

| Dataset | Parameter budget |
|---|---|
| ogbg-molhiv* | 500,000 |
| ogbg-molpcba | 2,000,000 |
| ogbn-arxiv | 500,000 |
| ogbn-products | 500,000 |
| mutag** | 500,000 |
| enzymes | 500,000 |
| proteins** | 500,000 |
| ptc-mr** | 500,000 |
| imdb-binary** | 500,000 |
| zinc | 100,000 |
| peptides-func | 2,000,000 |
| peptides-struct | 2,000,000 |

## C.5 Implementation details

We conduct our experiments using PyTorch Geometric as the underlying framework, running them on NVIDIA RTX A5000 GPUs. Detailed information on the selected hyperparameters for each dataset and layer configuration, along with instructions for reproducibility, can be found in our GitHub repository (`https://almogdavid.github.io/SSMA/` ). For ESAN [3] and VPA [38], we integrated our SSMA directly into the official implementation shared by the authors (ESAN, VPA). We performed hyperparameter search only within the space the authors used, without altering the training or evaluation protocols.

## C.6 Hyper parameter search

We use the Weights & Biases platform to perform hyperparameter searches (HPS), aiming to identify the optimal configuration for each dataset. For each configuration, we determine the maximum hidden dimension that fits within the predetermined parameter budget.

- **MLP compression strength**: We search for the optimal compression rate of the low-rank MLP compressor rate. We define the compression rate $\gamma$ as the ratio between the bottleneck rank and the inner aggregation representation dimension $m$. We perform a simple range search over `[0.1, 0.25, 0.5, 0.75, 1.0]`.

- **Neighbor selection method:** We search for the best neighbor selection method. We perform a simple range search over `[random, attention_slots]`.

- **Effective neighborhood size**: we search for the optimal neighborhood size $\kappa$. We perform a simple range search over `[2,3,...,CLIP(max_neighbors,7)]`.

# D Additional Experiments

## D.1 Comparison to variance preserving aggregation

We further evaluate our approach against a recently proposed method [39], we substitute the suggested aggregation technique with our own while retaining the original architecture and training protocol outlined in the work. We refer the reader to the original paper for an overview of architecture and training procedures. As illustrated in Table 6, our method demonstrates notable superiority over existing method without additional adjustments. Furthermore, optimizing hyperparameters and architecture selection has the potential for further enhancement.

Table 6: Test accuracy (higher is better). Shown is the mean ± STD of 10-fold cross-validation runs, VPA results are taken directly from [39], SSMA results are generated by us using the code provided in [39] without any architecture or training protocol modifications

| Module | IMDB-B | IMDB-M | RDT-B | RDT-M5K | COLLAB | MUTAG | PROTEINS | PTC | NCI1 |
|---|---|---|---|---|---|---|---|---|---|
| GCN + VPA | 71.7±3.9 | 46.7±3.5 | 85.5±2.3 | 54.8±2.4 | 73.7±1.7 | 76.1±9.6 | 73.9±4.8 | 61.3±5.9 | 79.0±1.8 |
| GCN + SSMA | **74.2±5.6** | **49.9±5.6** | **87.7±3.8** | **55.2±3.2** | **74.4±2.3** | **87.2±9.4** | **75.5±6.1** | **66.5±9.5** | **81.8±2.6** |
| GAT + VPA | 71.1±4.6 | 44.1±4.5 | 78.1±3.7 | 43.3±2.4 | 69.9±3.2 | 81.9±8.0 | 73.0±4.2 | 60.8±6.1 | 76.1±2.3 |
| GAT + SSMA | **78.6±5.8** | **50.5±4.8** | **82.6±5.8** | **50.5±4.8** | **76.9±2.1** | **88.3±11.5** | **76.6±5.4** | **65.7±11.9** | **81.8±7.9** |
| GIN + VPA | 72.0±4.4 | 48.7±5.2 | 89.0±1.9 | 56.1±3.0 | 73.5±1.5 | 86.7±4.4 | 73.2±4.8 | 60.1±5.8 | 81.2±2.1 |
| GIN + SSMA | **73.1±12.9** | **49.7±10.7** | **89.4±8.1** | **57.7±5.5** | **74.0±4.2** | **87.7±11.7** | **73.9±6.5** | **64.1±12.0** | **81.7±3.1** |
| Avg. improvment (%) | 5.19 | 7.80 | 2.93 | 6.74 | 3.88 | 7.85 | 2.68 | 7.32 | 3.88 |

## D.2 Comparison to Generalised f-Mean Aggregation

We conducted a further evaluation of our approach against a recently proposed aggregation function that parameterizes a function space encompassing all standard aggregators [25]. This aggregation was incorporated into our framework, and the experiments were performed using the identical setup described in Appendix C. For the new aggregation method, we employed the configuration specified by the authors in their experiments, as detailed in their repository). As shown in Appendix D.2, SSMA outperforms the proposed method. This demonstrates that even a method capable of learning a variety of aggregation functions experiences a relative decline in performance if it cannot effectively mix node features like SSMA. This underscores the critical importance of feature mixing in SSMA.

Table 7: Results for TU datasets [32] & ZINC [19] using the aggregation from [25] aggregation as a baseline. We report the TU datasets' accuracy mean and STD of a 10-fold cross-validation run. For the ZINC dataset, we report mean MAE and STD on the test set according to 5 distinct runs. [†] indicates reproduced results while [*] indicates the reported results from the relevant paper.

| Module | ENZYMES ↑ | PTC-MR ↑ | MUTAG ↑ | PROTEINS ↑ | IMDB-B ↑ | ZINC ↓ |
|---|---|---|---|---|---|---|
| GCN[†] [24] | 44.33±5.84 | 58.61±6.83 | 84.15±12.32 | 72.13±4.42 | 69.00±6.51 | 0.29±0.01 |
| GCN + SSMA | **54.83±7.55** | **62.29±9.33** | **89.79±6.71** | **76.28±3.19** | **75.2±2.9** | **0.280±0.02** |
| GAT[†] [43] | 50.17±7.60 | 59.16±7.97 | 83.25±10.53 | 73.75±4.5 | 68.10±6.49 | 0.39±0.33 |
| GAT + SSMA | **56.67±3.72** | **66.41±5.69** | **89.19±4.58** | **80.18±0.1** | **74.5±4.14** | **0.223±0.028** |
| GATv2[†] [6] | 51.67±8.05 | 57.43±7.49 | 77.05±12.75 | 70.96±5.47 | 71.8±5.09 | 0.26±0.01 |
| GATv2 + SSMA | **52.50±8.43** | **61.64±6.80** | **88.80±11.80** | **75.28±4.80** | **72.8±4.92** | **0.235±0.003** |
| GIN[†] [48] | 15.33±3.5 | 57.06±7.74 | 70.94±10.7 | 68.26±7.66 | 57.6±8.41 | 0.27±0.04 |
| GIN + SSMA | **51.69±8.04** | **61.28±9.23** | **90.51±6.97** | **75.19±4.73** | **74.1±5.02** | **0.222±0.003** |
| GraphGPS[†] [37] | 20.33±11.35 | 59.26±10.28 | 55.73±18.04 | 44.85±11.03 | 53.5±6.35 | 0.25±0.01 |
| GraphGPS + SSMA | **49.17±3.15** | **63.02±4.93** | **86.07±7.95** | **75.56±4.24** | **71.1±4.79** | **0.22±0.005** |
| Improvement (%) | 83.45 | 7.92 | 22.22 | 19.83 | 16.26 | 17.13 |

## D.3 Comparison to GraphGPS with Positionl-Encoding

Given the significant improvement in GraphGPS performance with positional encoding, we conducted additional experiments involving GraphGPS with positional encoding. The experiment details are provided in Appendix C. As shown in Appendix D.3, SSMA enhances GraphGPS performance even with positional encoding.

Table 8: Results for GraphGPS with positional encoding, the aggregation used for the baselines is Add. See Appendix C for more information.

| Dataset | GraphGPS | GraphGPS + SSMA | GraphGPS + PE | GraphGPS + + PE + SSMA |
|---|---|---|---|---|
| ENZYMES ↑ | 48.33±6.71 | **49.17±3.15** | 57.66±8.43 | **58.83±5.35** |
| PTC-MR ↑ | 61.41±6.91 | **63.02±4.93** | 59.58±5.34 | **64.76±5.72** |
| MUTAG ↑ | 79.91±10.23 | **86.07±7.95** | 90.34±7.78 | **91.37±5.76** |
| PROTEINS ↑ | 73.76±6.05 | **75.56±4.24** | 73.57±4.31 | **76.02±3.37** |
| IMDB-B ↑ | 69.6±5.54 | **71.1±4.79** | 70.9±4.6 | **72.5±4.68** |
| ZINC ↓ | 0.251±0.012 | **0.22±0.005** | 0.102±0.004 | **0.100±0.003** |
| PEPTIDES-F ↑ | 58.81±1.22 | **60.34±1.49** | 59.71±0.86 | **59.87±0.72** |
| PEPTIDES-S ↓ | 0.28±0.01 | **0.27±0.03** | 0.278±0.003 | **0.265±0.004** |
| OGBN-ARXIV ↑ | 63.87±0.68 | **66.71±0.73** | 53.53±1.98 | **62.85±2.4** |
| OGBN-PRODUCTS ↑ | 48.89±7.47 | **67.62±5.46** | 39.01±3.06 | **61.82±2.9** |
| OGBG-MOLHIV ↑ | 76.2±2.72 | **78.4±1.83** | 74.4±2.369 | **75.73±1.894** |
| OGBG-MOLPCBA ↑ | 0.19±0.01 | **0.22±0.01** | 0.196±0.006 | **0.199±0.006** |
| Improvement (%) | | 8.2 | | 8.64 |

# E   Ablation studies

## E.1   Neighbor selection method

In this experiment, we compare the strategy of selecting random neighbors for each node with our proposed soft-neighbor selection mechanism. There are two main reasons for this comparison. i) To demonstrate that SSMA can establish strong aggregation capabilities independently of the aggregation occurring in the attention slots. ii) To provide empirical justification for the proposed soft-neighbor selection mechanism.

To achieve this, we conducted ablation experiments on two different datasets:

1. "OGBN-Arxiv," a citation network where most nodes have a very low in-degree, while a few nodes have an extremely high in-degree.
2. "Proteins," a dataset with an in-degree distribution highly concentrated around the mean.

For each one of these datasets, we compared the test accuracy using both neighbor selection methods for a varying number of neighbors and types of MPGNN layers. The results are presented in Figure 5.

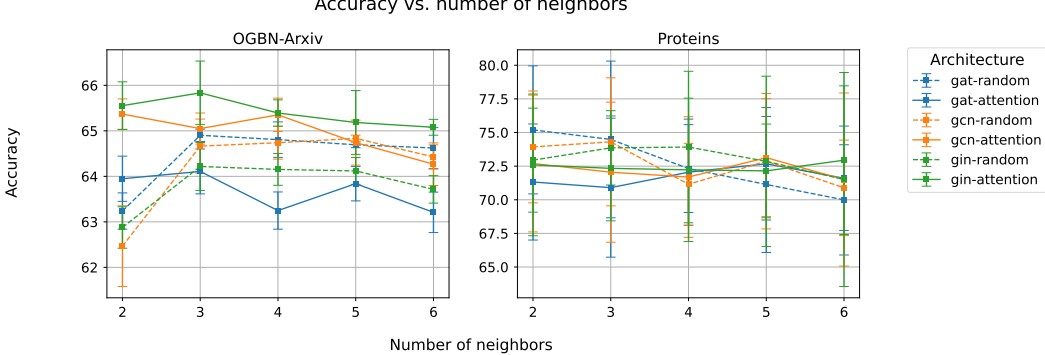

Figure 5: Comparison of the neighbor selection methods across different neighbor counts and MPGNN layer types on the "OGBN-Arxiv" and "Proteins" datasets.

The experiment results confirm that SSMA achieves strong performance even with the random neighbor selection and that the soft-neighbor selection works better in most cases. Interestingly, we observe that an increase in the number of neighbors does not necessarily correlate with improved performance, which was apparent most prominently in "OGBN-Arxiv."

Furthermore, we find that the GAT layer does not benefit from the proposed slot attention mechanism. This lack of improvement may be attributed to the intrinsic attention mechanism within the GAT architecture, rendering the additional attention mechanism redundant.

### E.2 Performance of SSMA under different budget constraints

In this experiment, we explore the performance of SSMA compared to sum-based aggregators across various hidden dimension sizes used by the MPGNN architectures. This investigation is motivated by two key objectives: i) To assess the relative gain of SSMA over sum-based aggregators in both the low-budget and high-budget regimes. ii) To analyze how the scaling behavior of sum-based aggregators compares to that of SSMA.

We conducted ablation studies on the "IMDB-B" and "MUTAG" datasets to achieve these goals. For each dataset, we measured the test accuracy using both sum-based aggregators and SSMA, varying the hidden dimensions and types of MPGNN layers. The results of these experiments are illustrated in Figure 6.

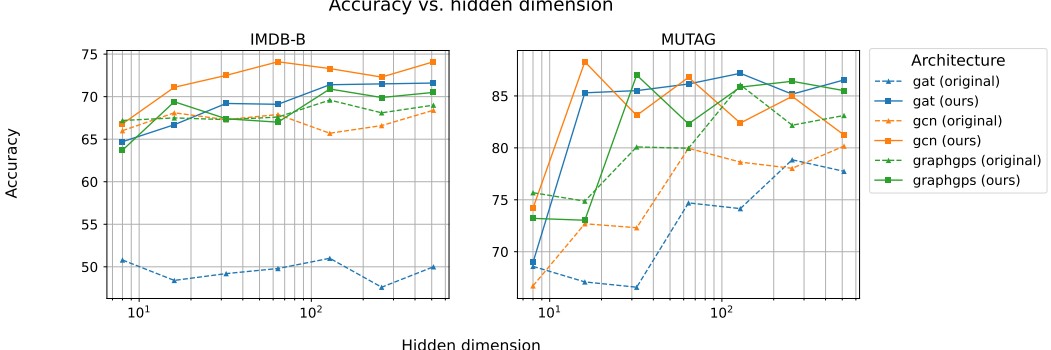

Figure 6: SSMA achieves peak performance with significantly lower hidden dimensions.

The experimental results clearly indicate that SSMA outperforms sum-based aggregators across all parameter regimes, showcasing its effectiveness in propagating relevant information for downstream tasks. Additionally, SSMA does not always benefit from higher dimensionality, reaching saturation much earlier than its counterpart aggregators. This further demonstrates the efficiency of SSMA.

### E.3 On the effectiveness of low-rank compressors

This experiment investigates the impact of low-rank MLP compression on the test performance of diverse architectures. As detailed in Section 4.4, low-rank compression significantly reduces learnable parameters while maintaining good expressive power. This allows for an intriguing trade-off: fewer parameters for a larger number of hidden units or slots. We evaluate this trade-off on the "OGBN-Arxiv" and "ZINC" datasets to understand its effect on performance.

The experiment results are presented in Figure 7.

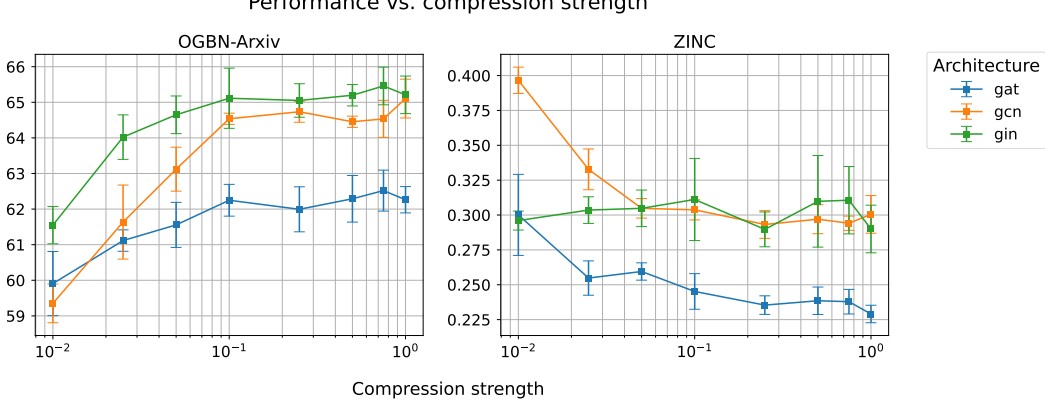

Figure 7: The performance under different compression strengths. SSMA can handle strong compression rates without losing much of its performance.

As demonstrated above, we can observe that SSMA exhibits resilience to strong compression, maintaining high performance. Notably, the best results are achieved at compression strengths significantly lower than one, demonstrating the expressive power of the MLP compressor even under significant compression and the efficiency of SSMA in propagating information.

### E.4 Learning the affine transformation in SSMA

To validate the selection of our affine transformation, we conducted an ablation study where we initialized the affine transformation with our proposed configuration and allowed the model to optimize it further. We used the ZINC dataset for this study and performed a brief hyperparameter search to identify the optimal configuration. The best results are presented in Table 9. The experiments show a slight improvement when the affine transformation is learned, but the gain is minimal, supporting the validity of our proposed affine transformation. Additionally, we examined the differences between the learned transformation and our proposed one. The learned transformation was similar to our proposal, with an average absolute difference of 0.023 ± 0.019, while the average norm of the affine weights is 0.026 ± 0.057.

Table 9: Learning the affine transformation on the ZINC dataset

| Layer | Learnable Affine | Constant Affine |
|-------|------------------|-----------------|
| GCN | **0.2822 ± 0.008** | 0.2836 ± 0.012 |
| GAT | **0.2278 ± 0.005** | 0.2323 ± 0.003 |
| GIN | **0.2210 ± 0.004** | 0.2260 ± 0.003 |

# F  Common aggregation functions benchmarks

In order to further show the effectiveness of SSMA we perform more benchmarks on other common aggregation functions. The results can be seen in the tables below.

Table 10: Results for TU datasets [32] & ZINC [19] using **LSTM** module as an aggregation as a baseline. We report the TU datasets' accuracy mean and STD of a 10-fold cross-validation run. For the ZINC dataset, we report mean MAE and STD on the test set according to 5 distinct runs. [†] indicates reproduced results while [*] indicates the reported results from the relevant paper.

| Module | ENZYMES ↑ | PTC-MR ↑ | MUTAG ↑ | PROTEINS ↑ | IMDB-B ↑ | ZINC ↓ |
|---|---|---|---|---|---|---|
| GCN[†] [24] | 37.5±8.21 | 58.79±5.23 | 78.85±21.38 | 66.58±7.85 | 54.2±7.38 | 0.32±0.04 |
| GCN + SSMA | **54.83±7.55** | **62.29±9.33** | **89.79±6.71** | **76.28±3.19** | **75.2±2.9** | **0.280±0.02** |
| GAT[†] [43] | 32.83±8.20 | 61.35±7.00 | 74.27±23.17 | 64.32±5.89 | 48.80±6.96 | 0.31±0.04 |
| GAT + SSMA | **56.67±3.72** | **66.41±5.69** | **89.19±4.58** | **80.18±0.1** | **74.5±4.14** | **0.223±0.028** |
| GATv2[†] [6] | 34.33±6.54 | 57.06±7.99 | 73.16±21.88 | 65.07±8.57 | 54.90±5.82 | 0.30±0.02 |
| GATv2 + SSMA | **52.50±8.43** | **61.64±6.80** | **88.80±11.80** | **75.28±4.80** | **72.8±4.92** | **0.235±0.003** |
| GIN[†] [48] | 37.83±7.16 | 58.41±7.97 | 78.72±23.88 | 71.07±5.61 | 50.70±8.21 | 0.25±0.03 |
| GIN + SSMA | **51.69±8.04** | **61.28±9.23** | **90.51±6.97** | **75.19±4.73** | **74.1±5.02** | **0.222±0.003** |
| GraphGPS[†] [37] | 29.17±10.37 | 58.17±5.37 | 73.38±22.01 | 63.03±14.57 | 49.90±4.98 | 0.32±0.02 |
| GraphGPS + SSMA | **49.17±3.15** | **63.02±4.93** | **86.07±7.95** | **75.56±4.24** | **71.1±4.79** | **0.22±0.005** |
| Improvement (%) | 55.39 | 7.09 | 17.52 | 16.11 | 42.53 | 20.93 |

Table 11: Results for TU datasets [32] & ZINC [19] using **max pooling** aggregation as a baseline. We report the TU datasets' accuracy mean and STD of a 10-fold cross-validation run. For the ZINC dataset, we report mean MAE and STD on the test set according to 5 distinct runs. [†] indicates reproduced results while [*] indicates the reported results from the relevant paper.

| Module | ENZYMES ↑ | PTC-MR ↑ | MUTAG ↑ | PROTEINS ↑ | IMDB-B ↑ | ZINC ↓ |
|---|---|---|---|---|---|---|
| GCN[†] [24] | 49.33±7.67 | 56.73±5.91 | 83.08±10.18 | 70.7±4.42 | 66.9±4.72 | 0.32±0.0 |
| GCN + SSMA | **54.83±7.55** | **62.29±9.33** | **89.79±6.71** | **76.28±3.19** | **75.2±2.9** | **0.280±0.02** |
| GAT[†] [43] | 45.83±5.29 | 58.87±7.42 | 80.85±9.84 | 71.43±4.23 | 66.5±6.65 | 0.27±0.01 |
| GAT + SSMA | **56.67±3.72** | **66.41±5.69** | **89.19±4.58** | **80.18±0.1** | **74.5±4.14** | **0.223±0.028** |
| GATv2[†] [6] | 47.17±7.07 | 60.26±8.38 | 77.44±10.48 | 71.42±6.09 | 65.6±4.09 | 0.28±0.01 |
| GATv2 + SSMA | **52.50±8.43** | **61.64±6.80** | **88.80±11.80** | **75.28±4.80** | **72.8±4.92** | **0.235±0.003** |
| GIN[†] [48] | 44.67±6.61 | 60.79±5.88 | 79.96±11.44 | 70.44±4.42 | 51.6±5.02 | 0.41±0.01 |
| GIN + SSMA | **51.69±8.04** | **61.28±9.23** | **90.51±6.97** | **75.19±4.73** | **74.1±5.02** | **0.222±0.003** |
| GraphGPS[†] [37] | 21.33±3.67 | 57.91±8.23 | 57.61±18.77 | 48.01±6.65 | 49.4±5.25 | 0.4±0.02 |
| GraphGPS + SSMA | **49.17±3.15** | **63.02±4.93** | **86.07±7.95** | **75.56±4.24** | **71.1±4.79** | **0.22±0.005** |
| Improvement (%) | 38.46 | 6.26 | 19.13 | 17.93 | 24.58 | 27.36 |

Table 12: Results for TU datasets [32] & ZINC [19] using **mean** aggregation as a baseline. We report the TU datasets' accuracy mean and STD of a 10-fold cross-validation run. For the ZINC dataset, we report mean MAE and STD on the test set according to 5 distinct runs. [†] indicates reproduced results while [*] indicates the reported results from the relevant paper.

| Module | ENZYMES ↑ | PTC-MR ↑ | MUTAG ↑ | PROTEINS ↑ | IMDB-B ↑ | ZINC ↓ |
|---|---|---|---|---|---|---|
| GCN[†] [24] | 45.83±9.69 | 54.88±6.07 | 86.07±7.03 | 73.21±4.07 | 70.7±4.06 | 0.31±0.01 |
| GCN + SSMA | **54.83±7.55** | **62.29±9.33** | **89.79±6.71** | **76.28±3.19** | **75.2±2.9** | **0.280±0.02** |
| GAT[†] [43] | 48.33±7.97 | 57.66±6.86 | 78.72±14.14 | 73.12±4.51 | 70.8±4.73 | 0.25±0.01 |
| GAT + SSMA | **56.67±3.72** | **66.41±5.69** | **89.19±4.58** | **80.18±0.1** | **74.5±4.14** | **0.223±0.028** |
| GATv2[†] [6] | 51.67±8.75 | 57.79±9.09 | 79.27±13.2 | 73.11±6.35 | 70.8±3.49 | 0.25±0.01 |
| GATv2 + SSMA | **52.50±8.43** | **61.64±6.80** | **88.80±11.80** | **75.28±4.80** | **72.8±4.92** | **0.235±0.003** |
| GIN[†] [48] | 43.67±8.85 | 56.05±5.32 | 72.26±11.06 | 74.46±4.79 | 50.0±5.29 | 0.4±0.01 |
| GIN + SSMA | **51.69±8.04** | **61.28±9.23** | **90.51±6.97** | **75.19±4.73** | **74.1±5.02** | **0.222±0.003** |
| GraphGPS[†] [37] | 29.83±8.48 | 58.79±5.58 | 61.15±21.44 | 68.1±5.48 | 48.2±4.94 | 0.39±0.0 |
| GraphGPS + SSMA | **49.17±3.15** | **63.02±4.93** | **86.07±7.95** | **75.56±4.24** | **71.1±4.79** | **0.22±0.005** |
| Improvement (%) | 24.43 | 10.37 | 19.13 | 5.75 | 22.02 | 22.91 |

Table 13: Results for TU datasets [32] & ZINC [19] using **min pooling** aggregation as a baseline. We report the TU datasets' accuracy mean and STD of a 10-fold cross-validation run. For the ZINC dataset, we report mean MAE and STD on the test set according to 5 distinct runs. [†] indicates reproduced results while [*] indicates the reported results from the relevant paper.

| Module | ENZYMES ↑ | PTC-MR ↑ | MUTAG ↑ | PROTEINS ↑ | IMDB-B ↑ | ZINC ↓ |
|---|---|---|---|---|---|---|
| GCN[†] [24] | 45.67±7.42 | 56.97±5.01 | 87.18±6.43 | 73.12±5.42 | 68.9±4.01 | 0.35±0.01 |
| GCN + SSMA | **54.83±7.55** | **62.29±9.33** | **89.79±6.71** | **76.28±3.19** | **75.2±2.9** | **0.280±0.02** |
| GAT[†] [43] | 46.33±9.96 | 56.53±8.98 | 80.68±9.22 | 71.60±4.12 | 65.8±3.99 | 0.27±0.01 |
| GAT + SSMA | **56.67±3.72** | **66.41±5.69** | **89.19±4.58** | **80.18±0.1** | **74.5±4.14** | **0.223±0.028** |
| GATv2[†] [6] | 41.5±8.11 | 57.72±7.09 | 82.91±9.26 | 70.43±4.21 | 64.9±4.98 | 0.27±0.01 |
| GATv2 + SSMA | **52.50±8.43** | **61.64±6.80** | **88.80±11.80** | **75.28±4.80** | **72.8±4.92** | **0.235±0.003** |
| GIN[†] [48] | 35.67±4.25 | 58.44±5.27 | 73.03±11.12 | 69.44±2.89 | 50.0±5.29 | 0.41±0.01 |
| GIN + SSMA | **51.69±8.04** | **61.28±9.23** | **90.51±6.97** | **75.19±4.73** | **74.1±5.02** | **0.222±0.003** |
| GraphGPS[†] [37] | 26.17±4.23 | 57.55±7.09 | 69.27±13.11 | 59.23±5.73 | 49.6±5.27 | 0.41±0.04 |
| GraphGPS + SSMA | **49.17±3.15** | **63.02±4.93** | **86.07±7.95** | **75.56±4.24** | **71.1±4.79** | **0.22±0.005** |
| Improvement (%) | 40.33 | 9.59 | 13.76 | 11.8 | 25.21 | 28.51 |

Table 14: Results for TU datasets [32] & ZINC [19] using **multiplication** aggregation as a baseline. We report the TU datasets' accuracy mean and STD of a 10-fold cross-validation run. For the ZINC dataset, we report mean MAE and STD on the test set according to 5 distinct runs. [†] indicates reproduced results while [*] indicates the reported results from the relevant paper.

| Module | ENZYMES ↑ | PTC-MR ↑ | MUTAG ↑ | PROTEINS ↑ | IMDB-B ↑ | ZINC ↓ |
|---|---|---|---|---|---|---|
| GCN[†] [24] | 29.67±7.11 | 56.55±7.57 | 77.91±10.23 | 60.57±7.61 | 65.4±4.4 | - |
| GCN + SSMA | **54.83±7.55** | **62.29±9.33** | **89.79±6.71** | **76.28±3.19** | **75.2±2.9** | **0.280±0.02** |
| GAT[†] [43] | 23.17±8.83 | 54.55±8.49 | 76.97±11.12 | 63.68±5.59 | 64.9±5.36 | 0.98±0.02 |
| GAT + SSMA | **56.67±3.72** | **66.41±5.69** | **89.19±4.58** | **80.18±0.1** | **74.5±4.14** | **0.223±0.028** |
| GATv2[†] [6] | 23.83±6.43 | 56.15±8.61 | 75.85±11.28 | 59.63±7.37 | 65.0±4.59 | 0.98±0.01 |
| GATv2 + SSMA | **52.50±8.43** | **61.64±6.80** | **88.80±11.80** | **75.28±4.80** | **72.8±4.92** | **0.235±0.003** |
| GIN[†] [48] | 22.67±5.84 | 55.08±7.75 | 73.12±14.44 | 69.35±5.24 | 46.8±4.08 | 1.5±0.04 |
| GIN + SSMA | **51.69±8.04** | **61.28±9.23** | **90.51±6.97** | **75.19±4.73** | **74.1±5.02** | **0.222±0.003** |
| GraphGPS[†] [37] | 22.33±7.63 | 58.58±8.1 | 73.5±12.35 | 52.33±10.08 | 51.8±4.59 | 0.88±1.42 |
| GraphGPS + SSMA | **49.17±3.15** | **63.02±4.93** | **86.07±7.95** | **75.56±4.24** | **71.1±4.79** | **0.22±0.005** |
| Improvement (%) | 119.58 | 12.1 | 17.81 | 26.18 | 27.47 | 82.69 |

