# OpenReview forum: "Sequential Signal Mixing Aggregation for Message Passing Graph Neural Networks"
_NeurIPS.cc/2024/Conference — NeurIPS 2024 poster_

### Official Review · Reviewer_4Dt8 · 2024-07-10

**Soundness:** 3
**Presentation:** 3
**Contribution:** 3
**Rating:** 7
**Confidence:** 4

**Summary:**

This paper focuses on the aggregation module in Message Passing Graph Neural Networks (MPGNNs).
It tackles the problem that sum-based aggregators, even though widely used, fail to 'mix' features belonging to distinct neighbors, preventing them from succeeding at the downstream tasks.
 Accordingly, the authors introduce a novel plug-and-play aggregation for MPGNNs, denoted as Sequential Signal Mixing Aggregation (SSMA).
SSMA treats the neighbor features as 2D discrete signals and sequentially convolves them with circular convolution by applying 2D Fast Fourier Transform (FFT).
The authors also propose several designs to guarantee the practical use of the proposed methods.
In the empirical experiments, SSMA successfully improved the performance of several MPGNNs.

**Strengths:**

1. This work has a good motivation and a solid mathematical foundation driving the proposed methods.
2. This paper is well-written.
3. The empirical performance can well support the statement of the paper.
4. According to the best of my knowledge, this is the first work to enhance the aggregation module via circular convolution.

**Weaknesses:**

1. Even though with good theoretical computational complexity analysis, it will be good to have the empirical run-time comparison between the MPGNN and MPGNN+SSMA.
2. According to my knowledge, there are many works using pre-computed PE with conditional aggregation to improve the distinguishability of the neighbors during the 'mixing' of them, (e.g., [1], GraphGPS). They require extra one-time computation on PE but will not increase the complexity of the model inference. There are no comparisons to the line of work. (I am not sure about the exact GraphGPS configuration in the experiments. I have some concerns about it which are listed in "questions")











- [1] Dwivedi, Vijay Prakash, et al. "Graph Neural Networks with Learnable Structural and Positional Representations." International Conference on Learning Representations.

**Questions:**

1. What is the configuration of GraphGPS in the experiment? As in the paper, several configurations are provided.
2. In eq (8), the number of separators $m=n+1$ depends on the neighbor sizes; $\mathbf{h}: \mathbb{R} \to \mathbb{R}^m$ is an affine map. How this affine map is determined? Is it learnable? In graph learning, the $n$ is usually assumed unknown and varying. (the same question applies to vector-feature cases as well)

2. Following up on the Weakness 2. The experiment on ZINC follows the 100k parameter setting as ESAN, while other baselines are reproduced. Compared to the 500K parameter setting in GraphGPS, there are interesting points. Regular MPNNs, e.g., GCN, GAT, GIN, PNA, do not show significant improvement on the 500K setting, while GraphGPS is significantly improved. Based on Table 3 and Table B.1 in GraphGPS, GatedGCN ($\sim 0.090$), GraphGPS($\sim0.070$), GraphGPS+NoPE ($\sim0.110$), GINE+RWSE($\sim0.070$).
    - I am interested to know the improvement of GRASS on GraphGPS on the 500K setting
    - Can GRASS improve MPNNs with PE?

**Limitations:**

The limitation on computational complexity is adequately addressed by the proposed technique.s
No potential negative societal impact aware.

---

> ### Author Rebuttal · Authors · 2024-08-03
>
> We are pleased that the reviewer recognized the novelty of our proposed method, its underlying mathematical foundations and the empirical performance supporting the statement of the paper. We thank the reviewer for appreciating the paper’s motivation and writing.
>
> ***
>
> We would like to address some concerns mentioned by the reviewer.
>
> > Even though with good theoretical computational complexity analysis, it will be good to have the empirical run-time comparison between the MPGNN and MPGNN+SSMA.
>
> We appreciate the reviewer’s suggestion and agree that such a comparison would demonstrate the efficiency of SSMA when integrated into MPGNNs. We conducted training and inference time comparisons, evaluating SSMA-augmented MPGNNs against PNA and GraphGPS. To ensure fair assessment we enforce the same hidden-dimension and report the time spent on a single convolutional layer. Please refer to Table 1 in the attached PDF.
>
> The results highlight the impressive trade-off of SSMA between down-stream performance and practical training and inference time complexities.
>
> ***
>
> > What is the configuration of GraphGPS in the experiment? As in the paper, several configurations are provided.
>
>
> As several configurations are provided in GraphGPS, we tried to focus on the most standard configuration possible - A GatedGCN as the message-passing network, standard multi-head attention with 8 heads without PE or SE. We used [PyG](https://pytorch-geometric.readthedocs.io/en/latest/generated/torch_geometric.nn.conv.GPSConv.html)’s implementation for the layer.
>
> ***
>
> > In eq (8), the number of separators $m=n+1$ depends on the neighbor sizes; $\boldsymbol{h}: \mathbb{R} \rightarrow \mathbb{R}^m$ is an affine map. How this affine map is determined? Is it learnable? In graph learning, the $n$ is usually assumed unknown and varying. (the same question applies to vector-feature cases as well)
>
> The affine map $\boldsymbol{h}$ In Eq. (8) is fixed and refers to the padded coefficients of each $p_i$ as given by Eq. (5) (please refer to lines 140-141). The same applies for $\boldsymbol{\Phi}$ in the vector case, in which the full form of the affine map is given in Appendix A.3, Eq. (39). While this is the (fixed) affine map which was used in the experiments, Appendix E.4 includes an ablation study that investigates this exact matter.
>
> Regarding the value of $n$, the provided number of separators is satisfactory as long as the number of neighbors is upper bounded by $n$. If there are fewer than $n$ neighbors, the high-power coefficients will vanish.
>
> ***
>
> > Following up on the Weakness 2. The experiment on ZINC follows the 100k parameter setting as ESAN, while other baselines are reproduced. Compared to the 500K parameter setting in GraphGPS, there are interesting points. Regular MPNNs, e.g., GCN, GAT, GIN, PNA, do not show significant improvement on the 500K setting, while GraphGPS is significantly improved. Based on Table 3 and Table B.1 in GraphGPS, GatedGCN ($\sim 0.090$), GraphGPS($\sim 0.070$), GraphGPS+NoPE ($\sim 0.110$), GINE+RWSE($\sim 0.070$). I am interested to know the improvement of GRASS on GraphGPS on the 500K setting. Can GRASS improve MPNNs with PE?
>
> Following the reviewer’s suggestion, we perform the following additional experiments on the ZINC regression benchmark:
> * Regarding scaling, we scale up different SSMA augmented architectures (both GCN,GIN and GraphGPS) to the 500K parameter regime.
> * Reagrding positional encoding (PE): we examine the effect of RWSE-20 positional encoding on SSMA-augmented GraphGPS.
>
> For the GraphGPS experiments in this context we used [PyG](https://github.com/pyg-team/pytorch_geometric/blob/master/examples/graph_gps.py)'s example code, carefully adopting the hyperparameters and optimization parameters used in the ZINC experiments in the GraphGPS paper.
> Please refer to Table 2 for the results.
>
> There are several insights:
> * SSMA consistently improves the effectiveness, even at a higher scale and with or without PE.
> * SSMA is well-behaved with positional-encoding, proving to be very effective even with 100K parameters.
> * Scaling does not seem to be valuable when considering classic MPNNs (in contrast to GraphGPS) which strengthens the reviewer's hypothesis.
>
> ***
>
> Thanks again for the review, let us know if you have any further questions!

---

> > ### Comment · Reviewer_4Dt8 · 2024-08-12
> >
> > Thank you for the rebuttal.
> >
> > It well addresses my concerns and questions.
> >
> > I will raise the score.

---

### Official Review · Reviewer_aDFM · 2024-07-12

**Soundness:** 3
**Presentation:** 2
**Contribution:** 2
**Rating:** 6
**Confidence:** 2

**Summary:**

The paper analyzes the discrepancy between the theoretical guarantees of sum-based aggregators and their practical performance, which makes more complex aggregators preferred in practice. They define the notion of neighbor-mixing to explain this gap, and propose a novel aggrgeation module, named SSMA, which builds upon deepsets polynomial.

**Strengths:**

The theoretical analysis is sound and rigorous.
The experimental section shows promising performance in practice, and I particularly appreciated that the authors have adjusted the number of parameters in the augmented model to match that of the original, to ensure a fair comparison.

**Weaknesses:**

The paper is hard to follow, and I think adding some intuition on the results would significantly strengthen the paper. For example, while I would have liked to understand the intuition why sum-based aggregators have small neighbor-mixing (which is very briefly touched in lines 116 to 118, but, in my opinion, not sufficiently to have any intuition, and it seems more like a statement that it is the case).
Section 4 is generally hard because, while it is supposed to be a way of constructing the model, it ends up being a theoretical digression until 4.3, where then the model becomes clear.

**Questions:**

Can you expand on the intuition of sum-based aggregators lacking mixing abilities?

**Limitations:**

Addressed

---

> ### Author Rebuttal · Authors · 2024-08-03
>
> We thank the reviewer for appreciating the theoretical analysis and the experimental section, particularly our methodological approach to ensure fair comparison.
>
> ***
>
> We would like to address your quoted concerns:
>
> > Can you expand on the intuition of sum-based aggregators lacking mixing abilities?
>
> Note that additional intuition on sum-based aggregators lacking mixing abilities is provided in subsection 4.3. Specifically, please refer to lines 185-187 and Eq. (14) which intuitively state that for sum-based aggregators the “mixing” is done only in the MLP.
> Regarding section 4, to further exemplify the construction of the generalized DeepSets polynomial we provide an additional figure illustrating a concrete instantiation of the polynomial for a specific neighbor set. Please refer to Figure 1 in the attached PDF.
>
> ***
>
> If the reviewer remains unsatisfied with the intuitive explanations, we welcome further clarification on what specific aspects need more elaboration.

---

> > ### Comment · Reviewer_aDFM · 2024-08-12
> >
> > I would like to thank the authors for the rebuttal. I will keep my score since it was already positive. I also encourage the authors to include Figure 1 in the appendix, and refer to it in the main paper.

---

### Official Review · Reviewer_QvY7 · 2024-07-13

**Soundness:** 2
**Presentation:** 3
**Contribution:** 3
**Rating:** 6
**Confidence:** 3

**Summary:**

This paper introduces Sequential Signal Mixing Aggregation (SSMA) for Message Passing Graph Neural Networks (MPGNNs), addressing the limitations of traditional sum-based aggregation methods. SSMA enhances the mixing of features from distinct neighbors by treating neighbor features as 2D discrete signals and applying sequential convolution.

**Strengths:**

Treating neighbor features as 2D discrete signals and using sequential convolution is innovative. This approach allows for more effective mixing of features from different neighbors, addressing a significant limitation in traditional sum-based aggregators. By revealing that sum-based aggregators cannot adequately "mix" neighbor features, the paper paves the way for more aggregation techniques. Introducing a convolution-based aggregation module is an advancement in the field.

The theoretical foundation of the proposed SSMA method is robust, with well-detailed mathematical formulations and proofs. The paper derives the key equations and supports them with solid theoretical arguments.

The practical aspects, including learnable affine transformations, low-rank compression, and normalization, are well thought out and enhance the applicability and stability of SSMA in real-world scenarios.

**Weaknesses:**

- Limited Comparisons: The paper only compares SSMA with one other aggregation method in the appendix. Including comparisons with more diverse and state-of-the-art aggregation methods would strengthen the evaluation and provide a more comprehensive understanding of SSMA's advantages and limitations. Recent advancements such as Generalized f-Mean Aggregation, Principal Neighbourhood Aggregation (PNA), Hybrid Aggregation for Heterogeneous Graph Neural Networks (HAGNN), Robust Graph Neural Networks via Unbiased Aggregation, and GNN-VPA: A Variance-Preserving Aggregation Strategy for Graph Neural Networks could provide valuable benchmarks for comparison​ (ar5iv)​​ (ar5iv)​​ (ar5iv)​​ (ar5iv)​​ (ar5iv)​​ (ar5iv)​​.

- Computational Complexity: The paper would benefit from discussing SSMA's computational complexity compared to other aggregation method  s. Currently, the paper only addresses SSMA's time complexity. It would be better if it included a comparison with the time complexities of similar aggregation methods from related work. This would provide a clearer context for evaluating SSMA's efficiency and practicality.

**Questions:**

- The authors conducted extensive tests combining different MPGNN architectures with SSMA in the experiments. However, why is there only a comparison of SSMA with one other aggregator in the appendix? Could the authors provide more experimental results comparing SSMA with other aggregators to demonstrate its advantages better?

- Could the authors provide direct training time comparisons between MPGNNs using SSMA and other aggregators? This would help to illustrate the time cost associated with adding the SSMA module more clearly.

- In Figure 1, SSMA is demonstrated, but the nodes u, v, and w appear more like a set rather than having a sequential relationship. Does "sequential" refer to the pipeline being sequential, or is there a sequential relationship among these graph nodes?

- Are there specific types of graphs or tasks where SSMA may not perform as well as other methods?

---

> ### Author Rebuttal · Authors · 2024-08-03
>
> We are grateful for the reviewer's recognition of convolution-based aggregations as an advancement in the field, effectively addressing a major limitation of traditional sum-based aggregators. We thank the reviewer for highlighting the robust theoretical foundation, and appreciating the practical aspects that enhance SSMA's applicability and stability in real-world scenarios.
>
> ***
> We would like to address your questions:
>
> > The authors conducted extensive tests combining different MPGNN architectures with SSMA in the experiments. However, why is there only a comparison of SSMA with one other aggregator in the appendix? Could the authors provide more experimental results comparing SSMA with other aggregators to demonstrate its advantages better?
>
> Thank you for this feedback which motivated us to broaden the experiments.
> Following your and reviewer's @vJN5 advice, we compared SSMA to a wider spectrum of aggregations including LSTM and Generalized f-Mean Aggregation[1]. Please refer to tables 4,5 in the attached PDF for the results.
>
>
> A comparison to PNA[2] already exists in the main paper (Tables 1 and 2) and a comparison to variance-preserving aggregation[3] (VPA) already exists appendix D.1.
> Although comparing our results with those from 'Robust Graph Neural Networks via Unbiased Aggregation' could have broadened our experiments, we were unfortunately unable to find a code base for this paper.
>
> ***
>
> > Could the authors provide direct training time comparisons between MPGNNs using SSMA and other aggregators? This would help to illustrate the time cost associated with adding the SSMA module more clearly.
>
> We appreciate the reviewer’s suggestion and agree that such a comparison would demonstrate the efficiency of SSMA when integrated into MPGNNs. We conducted training and inference time comparisons, evaluating SSMA-augmented MPGNNs against PNA and GraphGPS. To ensure fair assessment we enforce the same hidden-dimension and report the time spent on a single convolutional layer. Please refer to Table 1 in the attached PDF for the results.
>
>
> The results highlight the impressive trade-off of SSMA between down-stream performance and practical training and inference time complexities.
>
> ***
>
> > In Figure 1, SSMA is demonstrated, but the nodes u, v, and w appear more like a set rather than having a sequential relationship. Does "sequential" refer to the pipeline being sequential, or is there a sequential relationship among these graph nodes?
>
> The term 'sequential' in SSMA refers to the sequential convolution of the transformed features of neighboring nodes, as outlined in Theorems 4.1 and 4.4, which underlie the construction of SSMA. Figure 1 demonstrates that this sequential convolution can be practically computed in the Fourier domain.
>
> ***
>
> > Are there specific types of graphs or tasks where SSMA may not perform as well as other methods?
>
> While the “vanilla” version of SSMA may fail in dense neighborhoods due to the representation size scaling quadratically with the number of neighbors (as pointed out in lines 212-213, 269-270) the proposed neighbor selection mechanism (lines 212-220) accounts for such cases, as later demonstrated in the experimental section (lines 252-255).
> We haven’t identified other potential failure cases, which we leave for future research.
>
> ***
>
> Thanks again for the review!
>
> Let us know if you have any further questions and consider modifying your score if you are satisfied with our response.
>
> ***
>
> [1]: Generalised f-Mean Aggregation for Graph Neural Networks. NeurIPS 2023.
>
> [2]: Principal Neighbourhood Aggregation for Graph Nets. NeurIPS 2020.
>
> [3]:  GNN-VPA: A Variance-Preserving Aggregation Strategy for Graph Neural Networks. ICLR 2024 Tiny Paper.

---

> > ### Comment · Reviewer_QvY7 · 2024-08-13
> > **Official Comment by Reviewer QvY7**
> >
> > Thank you for the detailed response. I raise my score to 6.

---

### Official Review · Reviewer_vJN5 · 2024-07-19

**Soundness:** 2
**Presentation:** 3
**Contribution:** 2
**Rating:** 5
**Confidence:** 4

**Summary:**

This paper introduces Sequential Signal Mixing Aggregation (SSMA), a novel aggregation method for Message Passing Graph Neural Networks (MPGNNs). SSMA addresses the shortcomings of traditional sum-based aggregators, which often struggle to effectively "mix" features from distinct neighbors. By treating neighbor features as 2D discrete signals and sequentially convolving them, SSMA significantly enhances feature mixing from distinct neighbors. Experimental results show that integrating SSMA into existing MPGNN architectures markedly improves performance across various benchmarks, achieving state-of-the-art results.

**Strengths:**

1.	The explanation of the limitations of sum-based aggregators is compelling and insightful, offering a fresh perspective on the problem and effectively motivating the proposed method.
2.	SSMA introduces an innovative approach to feature aggregation in MPGNNs, which can be efficiently implemented and scaled to accommodate larger graphs.
3.	The experimental results on both node and graph-level tasks clearly demonstrate the effectiveness of the proposed method.

**Weaknesses:**

1.	Other aggregation methods, such as LSTM[1] and DIFFPOOL[2], also effectively mix neighbors’ features. These methods should be included in the related work and experimental comparisons.
2.	The inductive setting mentioned in the experimental setup section is missing from the experiments section.

[1] Inductive Representation Learning on Large Graphs. NeurIPS 2017.
[2] Hierarchical Graph Representation Learning with Differentiable Pooling. NeurIPS 2018

**Questions:**

Please refer to the weaknesses.

**Limitations:**

The limitations have been demonstrated in the paper.

---

> ### Author Rebuttal · Authors · 2024-08-03
>
> We thank the reviewer for finding the discussed limitations of sum-based aggregators compelling and insightful, endorsing SSMA’s innovative approach and appreciating the experimental results which demonstrate the effectiveness of SSMA.
>
> ***
>
> We would like to address your quoted concerns:
>
> > Other aggregation methods, such as LSTM[1] and DIFFPOOL[2], also effectively mix neighbors’ features. These methods should be included in the related work and experimental comparisons.
>
> Thank you very much for this feedback which motivated us to broaden the experiments.
>
> Following your and reviewer's @QvY7 suggestion, we compared SSMA to more advanced aggregation methods such as LSTM and Generalized f-Mean Aggregation[1], along the comparisons to PNA[2] and VPA[3] which already exist in the paper. Refer to tables 4,5 in the attached PDF for the results.
>
> While DiffPool[5] is not an aggregation method per-se but rather a technique to pool the graph nodes’, your input led us to create a dense version of SSMA, allowing it to be incorporated into DiffPool and other methods in this line of work (e.g MinCutPool [4]).
> In addition, we compared the original version of DiffPool with SSMA-augmented version of DiffPool on the TU datasets using the benchmark code provided by [PyG](https://github.com/pyg-team/pytorch_geometric/tree/master/benchmark/kernel).
> Refer to table 3 in the attached PDF for the results.
>
> ***
>
> > The inductive setting mentioned in the experimental setup section is missing from the experiments section.
>
>
> We point out that the experimental section includes both the inductive and transductive settings.
>  To provide further clarity, we present the distinction between the inductive and transductive benchmarks used in our experiments.
>
> Inductive benchmarks:
> ogbg-molhiv, ogbg-molpcba, mutag, enzymes, proteins, ptc-mr, imdb-binary, zinc, peptides-func, peptides-struct.
>
>
> Transductive benchmarks: ogbn-arxiv and ogbn-products.
>
> To further eliminate confusions regarding this topic, we refer the reviewer to table 3 in the appendix which contains additional details about the datasets.
>
> ***
>
> Thanks again for the review!
>
> Let us know if you have any further questions and consider modifying your score if you are satisfied with our response.
>
> ***
>
> [1]: Generalised f-Mean Aggregation for Graph Neural Networks. NeurIPS 2023.
>
> [2]: Principal Neighbourhood Aggregation for Graph Nets. NeurIPS 2020.
>
> [3]:  GNN-VPA: A Variance-Preserving Aggregation Strategy for Graph Neural Networks. ICLR 2024 Tiny Paper.
>
> [4]: Spectral Clustering with Graph Neural Networks for Graph Pooling. ICML 2020.
>
> [5]: Hierarchical Graph Representation Learning with Differentiable Pooling. NeurIPS 2018.

---

> > ### Author Response · Authors · 2024-08-12
> >
> > Dear Reviewers,
> >
> > We hope this message finds you well. In the most respectful way possible, we would greatly appreciate your acknowledgment of the responses we have provided. We understand that the concerns raised are grounded in important factual matters, and we believe that we have addressed each one directly with clear additional results and thorough, non-evasive discussions.
> >
> > Thank you very much for your time and consideration.

---

> > > ### Comment · Reviewer_vJN5 · 2024-08-13
> > > **Thank you**
> > >
> > > Thank you for the clarification and additional experimental proof to address my concerns. I raise my score to 5.

---

> > > ### Author Response · Authors · 2024-08-13
> > >
> > > Dear Reviewers,
> > >
> > > We are extremely grateful to you for reviewing our responses. We deeply appreciate your positive feedback and the scores you raised. We promise to include all additional clarifications, experiments, and figures in the camera-ready version.

---

### Author Rebuttal · Authors · 2024-08-07

We sincerely appreciate the reviewers for their valuable feedback on our paper.

We were pleased to hear that the reviewers found the explanation of the limitations of sum-based aggregators "compelling and insightful, offering a fresh perspective on the problem and effectively motivating the proposed method" (@vJN5). They also noted that "introducing a convolution-based aggregation module is an advancement in the field" (@QvY7). The significance and rigorous approach of the experimental section were recognized: "The experimental section shows promising performance in practice, and I particularly appreciated that the authors have adjusted the number of parameters in the augmented model to match that of the original, to ensure a fair comparison" (@aDFM). Additionally, the paper was praised for being well-written with solid theoretical foundations: "This work has a good motivation and a solid mathematical foundation driving the proposed methods" and "This paper is well-written" (@4Dt8).

***

The reviewers' suggestions encouraged us to expand our experiments and provide additional illustrations of our method.

The results of the requested experiments are detailed in the attached PDF file. Specifically:

* Table 1 compares the training and inference times of SSMA against other methods, as requested by reviewers (@QvY7,@4Dt8).
* Table 2 demonstrates the effects of positional encoding (PE) and scale on the effectiveness of SSMA as requested by reviewers (@4Dt8).
* Table 3 demonstrates the benefit of SSMA to DIffPool as requested by the reviewer (@vJN5).
* Figure 1 illustrates a specific realization of the generalized DeepSets polynomial compared to DeepSets motivated by the reviewer's comments (@aDFM).
* Table 4 includes additional experiments for the f-Mean aggregation as requested by the reviewer (@QvY7).
* Table 5 includes additional experiments for the LSTM aggregation, as requested by the reviewer (@vJN5).

***

We thank the reviewers for spending time reviewing our paper and encourage further discussion on any issues that may arise from the rebuttal. If our responses and the additional experiments satisfactorily address the reviewers' concerns, we would be grateful if they could consider modifying our score.

---

### Decision · Program_Chairs · 2024-09-25

**Decision:**

Accept (poster)

**Comment:**

This paper studies the aggregation method in graph neural networks. It founds that the commonly used sum-based aggregators cannot "mix" features of distinct neighbors. The authors then propose SSMA (short for Sequential Signal Mixing Aggregation) to tackle the limitation with respect to neighbor mixing. There are a few minor concerns remained after reading the paper and author-reviewer discussion:
* First, the structure of this paper made some parts hard to understand. For example, the authors claim that the mixing value for sum-based aggregation is small, which is more like a statement at that point. And more detailed explanation is deferred until section 4.3. Restructuring or a better pointer to the explanations would help.

* Second, it  might be helpful to briefly explain the intuition of definition and theorem before or after it, to further enhance the clarity for broad audience. For example, it would be helpful to explain the intuition of equation (1) by expanding "in that the mutual effect of perturbing the features ...". Similar treatments could be done for proposition 3.2 and most theoretical parts in section 4.2.

Despite these minor concerns, I find the paper well-motivated and the method interesting and novel. All reviewers unanimously believe that this paper is above the acceptance threshold. I recommend accepting this work.